# UNCERTAINTY MODELING FOR FINE-TUNED IMPLICIT FUNCTIONS

**Anna Susmelj**
ETH AI Center

**Mael Macuglia**
ETH Zürich

**Nataša Tagasovska**
Prescient/MLDD, Genentech

**Reto Sutter, Sebastiano Caprara**
Balgrist University Hospital, USZ

**Jean-Philippe Thiran**
LTS5, EPFL

**Ender Konukoglu**
ETH Zürich

## ABSTRACT

Implicit functions such as Neural Radiance Fields (NeRFs), occupancy networks, and signed distance functions (SDFs) have become pivotal in computer vision for reconstructing detailed object shapes from sparse views. Achieving optimal performance with these models can be challenging due to the extreme sparsity of inputs and distribution shifts induced by data corruptions. To this end, large, noise-free synthetic datasets can serve as shape priors to help models fill in gaps, but the resulting reconstructions must be approached with caution. Uncertainty estimation is crucial for assessing the quality of these reconstructions, particularly in identifying areas where the model is uncertain about the parts it has inferred from the prior. In this paper, we introduce Dropsembles, a novel method for uncertainty estimation in tuned implicit functions. We demonstrate the efficacy of our approach through a series of experiments, starting with toy examples and progressing to a real-world scenario. Specifically, we train a Convolutional Occupancy Network on synthetic anatomical data and test it on low-resolution MRI segmentations of the lumbar spine. Our results show that Dropsembles achieve the accuracy and calibration levels of deep ensembles but with significantly less computational cost.

## 1 INTRODUCTION

Recent advancements in neural implicit functions have facilitated their use for 3D object representations and applications in novel views synthesis in computer vision. Neural Radiance Fields (NeRFs) (Mildenhall et al., 2021) have gained recognition for their ability to accurately synthesize novel views of complex scenes and became an important tool in applications of photorealistic rendering (Martin-Brualla et al., 2021), such as virtual reality and augmented reality. Signed Distance Functions (SDFs) (Park et al., 2019a; Gropp et al., 2020) is shown to be particularly useful in scenarios where precise boundary details are crucial, like industrial design and robotics. Occupancy Networks (Mescheder et al., 2019b), which model shapes as a probabilistic grid of space occupancy, excel in handling topological variations, making them ideal for medical imaging (Amiranashvili et al., 2022) and animation. Advancing this concept, Convolutional Occupancy Networks (Peng et al., 2020) integrate convolutional networks to enhance spatial learning, proving effective in detailed architectural modeling and complex reconstructions.

In practice, achieving optimal performance with these methods often requires densely sampled input data and a high degree of similarity between the training data and the target object. However, such ideal conditions are rarely met in practical applications like augmented reality, virtual reality, autonomous driving, and medical contexts, where inputs are typically sparse and less precise (Truong et al., 2023; Müller et al., 2022a). For example, in medical applications, generating precise 3D representations from sparse inputs is particularly crucial for morphological analysis (Tóthová et al., 2020). Patient-specific anatomical modeling significantly enhances the assessment of a patient's condition and aids in devising customized treatment plans (Turella et al., 2021).

In addition to sparsity, real-world data usually suffers from noise and corruption that leads to distribution shifts with respect to the training data, and thus to significant performance degradation.

Occlusion, noise, truncation, and lack of depth measurements (Liao et al., 2023) greatly affect reconstruction from monocular observations. Noise in estimations of camera poses inevitably degrades reconstruction from sparse views (Truong et al., 2023; Zhang et al., 2022; Chen et al., 2021b). Under input sparsity and distribution shifts, in safety-critical applications, such as medical (Zou et al., 2023) or autonomous driving (Kiran et al., 2021), it is crucial that inferred information is transparently disclosed to the end user, as it may significantly influence the decision-making process. One approach to this end is to quantify uncertainty in the reconstruction. For example, Shen et al. (2021) derived a variational inference objective to model predictive distribution in NeRFs. However, similar results are yet to be established for more generic applications of implicit functions.

To tackle the challenges of sparse and corrupted inputs, we propose to leverage dense, noise-free synthetic data for pre-training. For instance, a pretrained encoder can be shared across both synthetic and real datasets, while a lightweight implicit decoder is initially trained on synthetic data and subsequently fine-tuned on the sparse, noisy real data. This process introduces two primary sources of uncertainty: from the encoder's latent representation and the fine-tuning of the decoder. In this paper, we focus on the latter and demonstrate that epistemic uncertainty is an important, yet understudied source of error in implicit functions, which significantly impacts the predictions quality. To the best of our knowledge, this type of uncertainty quantification in 3D reconstruction for fine-tuned neural implicit functions has not been addressed in the literature.

Monte Carlo (MC) dropout (Gal and Ghahramani, 2016) and deep ensembles (Lakshminarayanan et al., 2017) are two commonly used baselines for estimating uncertainty in computer vision applications (Kendall and Gal, 2017), which could be readily applied to our task. MC dropout is simple to integrate and computationally efficient during training. However, it often underestimates uncertainty and requires multiple forward passes during inference, increasing computational cost (Postels et al.; 2021). Deep ensembles involve training multiple neural networks independently and averaging their predictions, capturing a wider range of potential outputs. This method provides improved predictive performance and better-calibrated uncertainty estimates but is computationally costly and memory-intensive. This becomes especially demanding in our fine-tuning context, where each model in an ensemble needs to be trained on both datasets. Here we introduce *Dropsembles*[1], a method that creates ensembles based on the dropout technique. Dropsembles aim to moderate the computational demands associated with ensembles while attempting to maintain prediction accuracy and uncertainty calibration of deep ensembles. Combined with Elastic Weight Consolidation (EWC) (Kirkpatrick et al., 2016), Dropsembles is able to mitigate distribution shifts between source and target datasets.

**Contributions** This paper introduces several contributions to address the gap in modeling uncertainties in fine-tuned neural implicit functions:

- To the best of our knowledge, we are the first to model epistemic uncertainty in the implicit decoder during fine-tuning from synthetic data.
- To this end, we introduce Dropsembles (overview in Figure 1) to achieve the performance of vanilla ensembles with significantly reduced computation cost (subsection 4.1).
- We introduce EWC-inspired regularization for task-agnostic uncertainty adaptation to take into account the distribution shift in uncertainty modeling (subsection 4.2).
- We include a series of experiments to validate the proposed methods, in a controlled benchmark on synthetic data and in a real-world medical application. We demonstrate that it is possible to achieve high reconstruction quality and preserve patient-specific details when using synthetic data in the form of an anatomical atlas (section 5).

## 2 RELATED WORK

**Implicit shape modeling from sparse input** Much attention has recently been directed towards the problem of novel view synthesis from sparse views enhancing this task using NeRFs (Truong et al., 2023; Chen et al., 2021a; Deng et al., 2022; Jain et al., 2021; Kim et al., 2022; Niemeyer et al., 2022; Roessle et al., 2022; Lin et al., 2021; Oechsle et al., 2021), SDFs (Yu et al., 2022; Yariv et al., 2020),

---

[1]https://github.com/klanita/Dropsembles

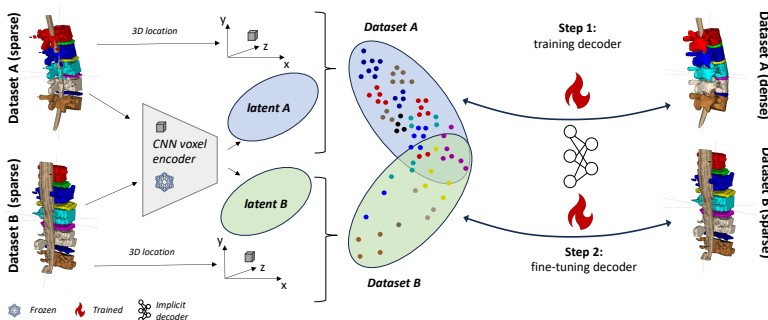

Figure 1: Occupancy network training with a dense prior and fine-tuning on a sparse dataset.

and occupancy networks (Niemeyer et al., 2020). Implicit functions also found their applications in medical imaging, particularly in 3D shape reconstruction from sparse MRI slices (Amiranashvili et al., 2022; 2024; Gatti et al., 2024; McGinnis et al., 2023; Alblas et al., 2023; Jacob et al., 2023). In this paper, we focus on occupancy networks due to their relevance in medical imaging applications but note that Dropsembles can be used with other neural implicit functions.

**Uncertainty modeling** Estimating uncertainty in deep models has attracted significant interest in recent years (Gal and Ghahramani, 2016; Lakshminarayanan et al., 2017; Gawlikowski et al., 2023). A remaining challenge here is uncertainty estimation under distribution shifts, which in most real-world scenarios is essential for guaranteeing the reliability and resilience of predictions when confronted with out-of-distribution (OOD) examples (Tran et al., 2022; Li et al., 2022; Wenzel et al., 2022). In 3D reconstruction, this is important since corruption and sparsity pattern in test samples often deviates from those in training data. While numerous methods have been devised to detect OOD cases (Malinin and Gales, 2018; Tagasovska and Lopez-Paz, 2019; Ren et al., 2019; Charpentier et al., 2020; Osband et al., 2023) or enhance accuracy in unobserved domains (Liu et al., 2021; Hendrycks et al., 2021), the adaptation of uncertainty estimates has been relatively unexplored. In a comprehensive benchmark study, Wenzel et al. (2022) find that from all metrics considered, calibration transfers worst, meaning that models that are well calibrated on the training data are not necessarily well calibrated on OOD data. In an era dominated by LLMs and foundation models, addressing this issue becomes paramount, particularly given the tendency of such models for overconfidence as a result of fine-tuning (Guo et al., 2017a; Dodge et al., 2020). To overcome such issues, (Balabanov and Linander, 2024) introduces a method for estimating uncertainty in fine-tuned LLMs using Low-Rank Adaptation. In computer vision, (Wortsman et al., 2022) highlights that fine-tuning comes at the cost of robustness, and address this issues by ensembling the weights of the zero-shot and fine-tuned models. (Lu and Koniusz, 2022) propose uncertainty estimation in one-shot object detection, particularly focusing on fine-tuned models.

Several works focused on modeling uncertainty in the encoder of neural implicit representations. (Shen et al., 2022) models uncertainty in the color and density output of a scene-level neural representation with conditional normalizing flows. Liao et al. (2023) uses this method in robotics applications. Liao and Waslander (2024) estimates uncertainty for neural object representation from monocular images by propagating it from image space first to latent space, and consequently to 3D object shape. Yang et al. (2023) focused on modeling aleatoric uncertainty in occupancy networks. Dropsembles can complement all of the above approaches by modeling the epistemic uncertainty in the weights of implicit functions throughout fine-tuning, rendering them more robust to distribution shifts. By doing so, it improves both the performance and reliability of methods building on neural implicit representations.

## 3 BACKGROUND

### 3.1 IMPLICIT SHAPE REPRESENTATIONS

Implicit functions are widely used in computer vision to represent complex shapes. Assuming a continuous encoding of the object domain $\mathcal{X}$, an SDF for a surface $S$ encoding a shape is defined as

$f : \mathcal{X} \rightarrow \mathbb{R}$ where: $f(x) < 0 \Leftrightarrow x$ "inside" $S \quad f(x) = 0 \Leftrightarrow x \in S$. Another popular way to represent a shape object is an occupancy function $g : \mathcal{X} \rightarrow \{0, 1\}$ defined as $g(x) = \mathbb{I}_{\text{Shape Object}}(x)$ where $\mathbb{I}$ corresponds to the indicator function. The primary advantage of both approaches lies in their ability to accurately model complex shapes from sparse observations using deep neural networks. This capability underpins methods such as "DeepSDF" (Park et al., 2019b) and "Occupancy Network" (Mescheder et al., 2019a), which approximate functions by non-linearly regressing observations to the surface encoding the shape. To facilitate learning from training sets, these techniques are often employed in conjunction with latent representations to capture diverse shape variations and at inference time yield continuous shape representations. The latent representations can be trained directly from data using autoencoders (Mescheder et al., 2019a) or autodecoders (Park et al., 2019b). Recent research has also explored using large pretrained models like CLIP to obtain latent representations (Cheng et al., 2023). While the specific choice of the latent encoder is beyond the scope of this paper, we will refer to it as an "oracle".

Both methods can be described with the same formulation. We consider a dataset $\mathcal{D} := \left\{ \{x_j^i, y_j^i\}_{j \in [K]}, Z^i \right\}_{i \in [N]}$, where $N$ is the number of images, $K$ is the number of available pixels per image , $Z^i$ is a latent representation of image $i$ given by the "oracle", $x_j^i \in \mathcal{X} \subseteq \mathbb{R}^3$ is the position corresponding to the observed value $y_j^i$, which are obtained through an SDF or voxel occupancy following the definitions above. In both settings, the regression function is a neural network $f_\theta(x, Z), \theta \in \Theta_{NN}$. At training, networks are trained using the following optimization objective.

$$\hat{R}_{\mathcal{D}}(\theta) = \frac{1}{N \cdot K} \sum_{j \in [K] \,\wedge\, i \in [N]} l(f_\theta(x_j^i, Z^i), y_j^i) \tag{1}$$

Here, the loss function $l$ depends on the method. For occupancy networks, $l$ is the binary cross-entropy, reflecting the binary nature of occupancy, while it is the $L_2$ norm for SDFs, aligning with the continuous nature of the distance function.

## 3.2 UNCERTAINTY MODELING

**Deep Ensembles** Deep Ensembles (Lakshminarayanan et al., 2017) apply ensemble methods to neural networks (Lee et al., 2020; Breiman, 2001; Schapire, 1990; Saberian and Vasconcelos, 2010). Multiple independent networks are trained to determine a set of optimal weights $\{\hat{\theta}_i\}_{i \in [M]}$ with the subscript $i$ denoting different networks. Ensembles mitigate the risk of selecting a single set of weights, which may not yield good results particularly when training data consist of fewer samples relative to the size of the parameter space. Instead, multiple solutions with comparable accuracy are determined, allowing the ensemble to average outputs and minimize the selection risk. Training of deep ensembles is associated with high computational demands due to training multiple networks.

**Dropout** Dropout (Srivastava et al., 2014) is a regularization technique that aims to reduce overfitting by randomly omitting subsets of features during each training step. By "dropping out" (i.e., setting to zero) a subset of activations within a network layer, it diminishes the network's reliance on specific neurons, encouraging the development of more robust features. Gal and Ghahramani (Gal and Ghahramani, 2016) demonstrated that dropout can also be interpreted from a Bayesian perspective and applied toward modeling uncertainty. Dropout at test time (referred to as Monte Carlo Dropout) is shown to perform approximate Bayesian inference, essentially through using randomness in dropout configurations for uncertainty modeling.

## 3.3 ELASTIC WEIGHT CONSOLIDATION

Elastic Weight Consolidation (EWC) is a regularization technique introduced in continual learning to address catastrophic forgetting (Kirkpatrick et al., 2016). The underlying principle is to protect parameters crucial for previous tasks while learning new ones. Assume two distinct tasks, $A$ and $B$, with their respective datasets $\mathcal{D}_A$ and $\mathcal{D}_B$, where $\mathcal{D}_A \cap \mathcal{D}_B = \emptyset$. The tasks are learned sequentially without access to previous tasks' datasets. First, task $A$ is learned by training a neural network on $\mathcal{D}_A$, resulting in a set of optimal weights $\hat{\theta}_A$. When learning task $B$ using dataset $\mathcal{D}_B$, EWC regularizes the weights so they remain within a region in the parameter space that led to good accuracy for task $A$.

Next, during learning for task $B$, approximate posterior of weights obtained during learning for $A$ constrains the optimization. A Gaussian approximation to $\log p(\theta|\mathcal{D}_A)$ (see App. A for further details), serves as the regularizer

$$\hat{\theta}_B = \arg\min_{\theta} \hat{R}_{\mathcal{D}_B}(\theta) + \lambda(\theta - \hat{\theta}_A)^T F(\hat{\theta}_A)(\theta - \hat{\theta}_A) \tag{2}$$

where $\hat{R}_{\mathcal{D}_B}(\theta)$ corresponds to the likelihood term $\log p(\mathcal{D}_B|\theta)$, $\lambda$ is a hyperparameter, $F$ is the diagonal of Fisher information matrix. Details can be found in Appendix A

## 4 METHODS

We focus on shape reconstruction from sparsely sampled and corrupted inputs using occupancy networks. While the proposed method can be applied to both SDF and occupancy networks, we focus on the latter for demonstration. Given the sparse and corrupted nature of our input, we train an occupancy network with high-quality data and then fine-tune it on a test sample. Through fine-tuning, we expect the model to adapt to the input while transferring the prior information captured in training.

Assume access to datasets $\mathcal{D}_A$ and $\mathcal{D}_B$, defined in Section 3.1, with the number of points per image $K_A$ and $K_B$, and the number of images $N_A$ and $N_B$. The dataset $\mathcal{D}_A$ is assumed to be high quality and "dense", while the dataset $\mathcal{D}_B$ is "sparse" in terms of the number of points observed per image and potentially contains corruptions in individual images, i.e., $K_A \gg K_B$. Additionally, it is assumed that the number of images in $\mathcal{D}_A$ is greater than in $\mathcal{D}_B$, i.e., $N_A > N_B$ (in our experiments, we consider the case of a single image $N_B = 1$). The two datasets can only be accessed successively and not simultaneously; that is, $\mathcal{D}_A$ is accessed first, followed by $\mathcal{D}_B$, without further access to $\mathcal{D}_A$. Note that the latent representation vectors $Z^i$ are assumed to be obtained by the same learning oracle for both datasets $\mathcal{D}_A$ and $\mathcal{D}_B$ described in Section 3.1. Without loss of generality, in this paper, we consider the latent map $\mathcal{L} : \mathcal{X} \to \mathcal{Z}$ to be a "frozen" encoder pretrained on dataset $\mathcal{D}_A$.

The underlying parametric function class of the model is assumed to be a neural network $f_\theta :$ $\mathcal{X} \times \mathcal{Z} \to \mathcal{Y}$ approximating a regression function as described in Section 3.1. The parameter $\theta$ represents the set of weight matrices of the network i.e $\theta := \{W_i\}_{i \in [L]}$ where $L$ corresponds to the number of layers. The output space $\mathcal{Y}$ corresponds to $[0, 1]$ for occupancy networks and $\mathbb{R}$ for signed distance functions. It is important to note that the versatility of the network model in terms of its output space is tailored to the specific modeling task. However, this flexibility does not limit the proposed method, which remains versatile across different tasks.

The procedures outlined in Sections 4.1 and 4.2 involve two stages of training: initially on dataset $\mathcal{D}_A$, denoted as Task $A$, followed by training on dataset $\mathcal{D}_B$, referred to as Task $B$. The primary objective is to achieve high prediction accuracy in the second stage, while acknowledging the inherent challenges posed by the sparsity and smaller size of $\mathcal{D}_B$ relative to $\mathcal{D}_A$.

### 4.1 DROPSEMBLES

Given the sparsity and corrupted nature of test samples, adaptation of the prior model is prone to uncertainties. Here we introduce Dropsembles, a technique that combines benefits of both dropout and deep ensembles, to capture and quantify this uncertainty. This approach aims to moderate the computational demands associated with ensembles while attempting to maintain reasonable prediction accuracy. Although ensembles are known for their reliable predictions, they are resource-intensive. In real-world applications, involving large training sets, this cost becomes a hindering factor.

In contrast, dropout involves training only a single model instance. Applying dropout to a neural network involves sampling a "thinned" network, effectively the same as applying a binary mask to the weights. However, unlike ensembles where each model is trained independently, dropout results in networks that are not independent; they share weights. Thus, training a neural network with dropout is akin to simultaneously training a collection of $2^n$ thinned networks ($n$ - number of weights), all sharing a substantial portion of their weights. However, often this comes at a price of less accurate predictions.

In Dropsembles, the model is first trained with dropout on dataset $\mathcal{D}_A$. Subsequently, $M$ thinned network $\{f_{\theta_m}\}_{m \in [M]}$ instances are generated by sampling binary masks. Each of these "thinned"

instances is then fine-tuned on dataset $\mathcal{D}_B$ *independently*, effectively creating an ensemble of "thinned" networks initialized with correlated weights. For inference and uncertainty estimations, this ensemble is treated as a uniformly weighted mixture model, and the predictions are aggregated in the same manner as traditional ensembles: $p(y|x) = \frac{1}{M} \sum_{m=1}^{M} p_{\theta_m}(y|x, \theta_m)$. Although ensembles typically benefit from networks being large and independent, our experiments show that this relaxation of independence does not significantly diminish prediction performance metrics or expected calibration error. The overall training procedure is summarized in Algorithm 1.

In the context of implicit functions, networks are optimized according to the objective defined in Equation (1). However, it is important to emphasize that Dropsembles is a versatile framework that can be adapted to train with any objective necessary for a specific task.

---

**Algorithm 1** Dropsembles

                                                                             $\triangleright$ *Task A*

**Require:** $: \mathcal{D}_A$
1:   $\hat{p}(\theta|\mathcal{D}_A) \leftarrow$ Train $f_\theta$ on $\mathcal{D}_A$ with dropout

                                                                             $\triangleright$ *Task B*

**Require:** $: \mathcal{D}_B, \hat{p}(\theta|\mathcal{D}_A)$
2:   **for** $m = 1$ to $M$ **do**
3:      $\theta_{init}^m \leftarrow$ Sample a thinned network initialized from $\hat{p}(\theta|\mathcal{D}_A)$
4:      $\hat{\theta}^m \leftarrow \arg\min_\theta \hat{R}_{\mathcal{D}_B}(\theta)$                             $\triangleright$ Train thinned network on $\mathcal{D}_B$
5:   **end for**
6:   Obtain predictions and uncertainty estimates $\leftarrow$ Ensemble $\{\hat{\theta}^m\}_{m\in[M]}$

---

### 4.2 ELASTIC WEIGHT CONSOLIDATION REGULARIZATION FOR IMPLICIT SHAPE MODELING

In the basic version of optimization described above, networks start their training initialized from the learned posterior of dataset $\mathcal{D}_A$. However, there is no guarantee that during fine-tuning the network will not diverge arbitrarily from initial weights. This is particularly problematic for implicit shape modeling, where it is essential to retain information from a large and dense prior dataset $\mathcal{D}_A$ while adapting to sparse and noisy data $\mathcal{D}_B$ to avoid overfitting. To address this concern, we borrow developments from the continual learning literature, as they seamlessly fit into the framework described above.

In particular, when fine-tuning individual instances of thinned networks on dataset $\mathcal{D}_B$, we can apply the same reasoning to each network instance as described in EWC. Thus, the learning objective for part B is replaced by the objective described in (Equation 2), which includes an additional regularization term.

## 5 EXPERIMENTAL RESULTS

Our objective is to provide trustworthy predictions on dataset B without hurting performance. We do so by modeling the uncertainty of the weights of a fine-tuned model. In standard prediction tasks, models are evaluated with respect to accuracy and Expected Calibration Error (ECE). However, our specific setup (reconstruction from sparse views) calls for additional evaluation metrics suitable to computer vision tasks, such as Dice Score Coefficient (DSC) and Hausdorff distance. Besides ECE, we include reliability diagrams (Guo et al., 2017b) given our preference for a more conservative modeling approach. Details on evaluation metrics can be found in Appendix B.

All models were evaluated using four network instances. Training details are provided in Appendix C. The optimal regularization parameter for EWC was determined through an ablation study presented in Appendix D. Additional ablation results on the number of network instances are included in Appendix E.

### 5.1 CLASSIFICATION UNDER DISTRIBUTION SHIFT

**Toy dataset**   We first demonstrate our method on a toy data set for binary classification. We generated two-dimensional datasets A and B with a sinusoidal decision boundary and Gaussian noise. A

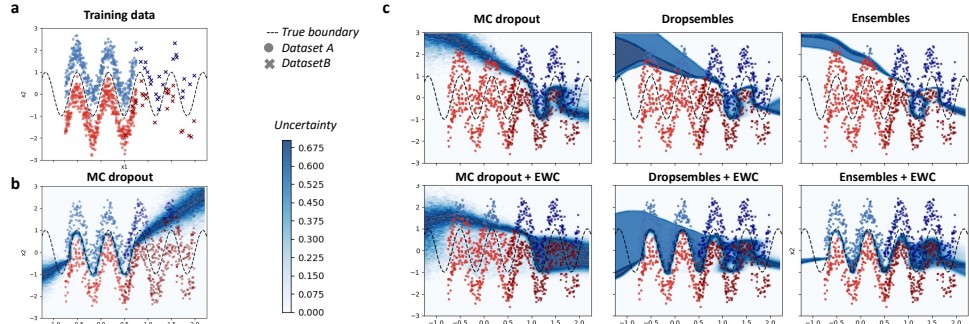

Figure 2: Toy classification example. a) Training data for binary classification task ("red" vs "blue") from datasets A (points, light) and B (crosses, dark) b) MC dropout trained only on Dataset A. c) Comparison of methods fine-tuned on Dataset B. Points are colored by the predicted class. EWC consistently improves both accuracy and uncertainty estimates on both A and B datasets.

moderate distribution shift was modeled between the datasets by adjusting the support values. We created 1000 train samples for dataset A, and 50 train samples for dataset B. All metrics in Table 1 were averaged across 3 random seeds. From the evaluations (Table 1, Figure 2), it is apparent that EWC improves the performance of Ensembles and Dropsembles on both datasets.

Table 1: Comparison of methods fine-tuned on dataset B for toy classification, MNIST and ShapeNet reconstruction experiments. Values are shown in percentages [%] for all metrics.

| | Toy classification | | | | MNIST | | ShapeNet | |
|---|---|---|---|---|---|---|---|---|
| Method | Acc-A ↑ | ECE-A ↓ | Acc-B ↑ | ECE-B ↓ | DSC ↑ | ECE ↓ | mDSC ↑ | ECE ↓ |
| MCdropout | $59.0 \pm 5.3$ | $40.0 \pm 4.3$ | $90.5 \pm 0.6$ | $11.5 \pm 1.7$ | $54.8 \pm 6.9$ | $8.6 \pm 2.0$ | $82.8 \pm 2.2$ | $19.31 \pm 3.97$ |
| Dropsembles | $59.2 \pm 4.3$ | $39.8 \pm 4.3$ | $90.5 \pm 0.6$ | $9.2 \pm 1.3$ | $64.1 \pm 4.8$ | $6.1 \pm 2.7$ | $83.5 \pm 2.1$ | $19.51 \pm 4.64$ |
| Ensembles | $56.5 \pm 2.4$ | $42.5 \pm 1.7$ | $90.0 \pm 0.0$ | $8.5 \pm 0.6$ | $62.6 \pm 5.1$ | $\underline{5.8} \pm 2.3$ | $82.6 \pm 2.2$ | $25.88 \pm 3.27$ |
| MCdropout + EWC | $66.8 \pm 5.6$ | $33.0 \pm 4.8$ | $86.5 \pm 2.1$ | $10.5 \pm 1.9$ | $55.7 \pm 7.2$ | $8.4 \pm 2.2$ | $\underline{83.9} \pm 2.4$ | $\mathbf{16.96} \pm 3.81$ |
| Dropsembles + EWC | $\mathbf{96.2} \pm 4.2$ | $\mathbf{5.8} \pm 5.3$ | $\underline{93.5} \pm 3.8$ | $\underline{7.0} \pm 1.4$ | $\mathbf{70.3} \pm 1.7$ | $6.1 \pm 1.9$ | $\mathbf{84.3} \pm 2.1$ | $\underline{17.67} \pm 4.45$ |
| Ensembles + EWC | $\underline{95.8} \pm 2.5$ | $\underline{7.5} \pm 2.4$ | $\mathbf{95.5} \pm 1.3$ | $\mathbf{5.5} \pm 1.0$ | $\underline{69.3} \pm 3.8$ | $\mathbf{5.2} \pm 2.0$ | $83.4 \pm 1.9$ | $24.59 \pm 2.98$ |

## 5.2 IMPLICIT SHAPE MODELING

**MNIST digit reconstruction**    In our next experiment, we explore the reconstruction from sparse inputs using the MNIST dataset. To this end, we converted the images into binary masks through thresholding. To simulate sparse input conditions, we applied a grid mask to the images, masking out every third row and column. This masking strategy was consistently applied across both datasets A and B. For dataset A, we utilized all images for a single digit "7" from the MNIST training set.

To introduce a moderate distribution shift and mimic common real-world dataset corruptions, we rotated the images in dataset B by 15 degrees and obscured approximately 15 percent of the original pixels. In this section we are focusing on shape reconstruction, therefore "fine-tuning" and "testing" are applied per image and not per dataset. Given that the occupancy network is trained at the pixel level, each of these images effectively constitutes an individual "dataset B". We randomly selected 20 distinct images from the MNIST test split, which would give us 20 different variations of "dataset B". Fine-tuning and evaluating the whole test split of MNIST dataset would be computationally demanding, as each occupancy network is fine-tuned on an individual image.

Our occupancy network comprises an 8-layer MLP, designed to process the latent representation along with the 2-dimensional coordinates of each pixel, thereby facilitating pixel-wise predictions. For configurations utilizing dropout, we incorporated a dropout layer with a probability of $p = 0.3$ following each linear layer in the network.

We demonstrate qualitative predictions and associated uncertainties for a single-image example in Figure 3. Visual inspection of the uncertainty estimates reveals that EWC-regularized ensembles exhibit the desired behavior: they not only deliver accurate predictions but also provide conservative

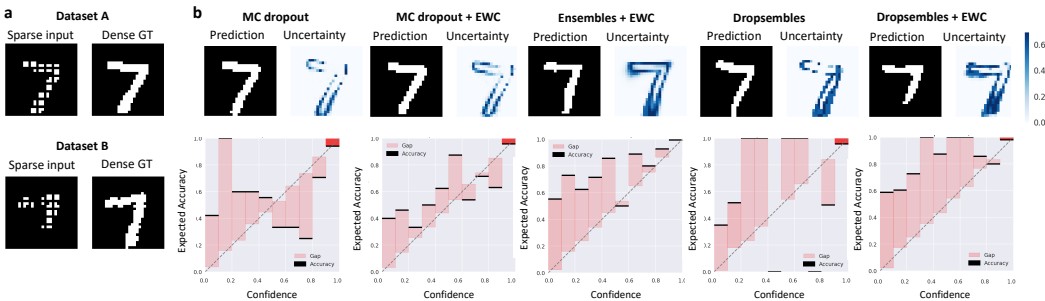

Figure 3: Corrupted MNIST reconstruction example. a) Example of training images. b) Comparison of fine-tuned methods on dataset B. A "perfectly calibrated" method would have reliability diagrams aligned on the diagonal. A "good conservative" method would have all bars above the diagonal.

uncertainty estimates, particularly noting high uncertainty in regions with data corruption. Reliability diagrams, along with quantitative evaluations presented in Table 1, confirm that elastic regularization enhances the performance across all methods. Notably, EWC-regularized Dropsembles achieve performance comparable to that of the Ensembles but with significantly reduced resource usage.

**ShapeNet** We demonstrate the applicability of Dropsembles for modeling uncertainty in SDFs on a commonly used ShapeNet dataset. For dataset A, we utilized 1,000 randomly selected airplane shapes. For dataset B, we selected 10 unseen airplane shapes and introduced a distribution shift by applying random erosion as corruption. Encoder inputs were provided as occupancy, with the target SDF estimated from the original meshes. For the corrupted dataset, the SDF was derived by converting the corrupted occupancies into a mesh and then estimating the SDF. The decoder follows the DeepSDF architecture with a dropout probability of $p = 0.2$ where applicable. Quantitative results in Table 1 demonstrate that Dropsembles outperform other methods in terms of DSC, with EWC regularization providing further improvements in both DSC and ECE metrics across all methods. Although MC dropout achieves the lowest ECE score, it fails to preserve shape smoothness when used for uncertainty modeling rather than regularization, as highlighted by additional qualitative results in Appendix Figure 11.

Recently, non-ReLU activation functions have gained popularity in neural implicit architectures (Sitzmann et al., 2020; Tancik et al., 2020; Müller et al., 2022b; Mehta et al., 2021; Dupont et al., 2022). In Appendix Table 4, we demonstrate that Dropsembles can be effectively combined with periodic activation functions without any loss in performance.

**Lumbar spine** In this section, we adapt the data preparation procedure from (Turella et al., 2021) but replace the high-quality CT dataset with a synthetic dataset of anatomical shapes. As an anatomical shape prior, we utilize a rigged anatomical model from "TurboSquid". To model patient-specific variability and variations in poses during MRI acquisition, we generated 94 rigged deformations. Point-cloud models from the atlas were converted to voxels at 256 voxel resolution in order to correspond to the resolution of the target MRI dataset. We further applied random elastic deformations to mimic patient-specific shape variability. We created a paired "sparse" - "dense" dataset by selecting a consistent set of 17–21 sagittal slices and additionally applied two iterations of connected erosions to simulate patient-specific automatic MR segmentations (Turella et al., 2021). A bicubic upsampling was employed on the "sparse" inputs to adapt them for the encoder. We trained ReconNet (Turella et al., 2021) on the entire training split of the atlas dataset to obtain an encoder, which we subsequently kept frozen. For the implicit decoder, we employed only one of the rigged samples to accurately quantify performance on synthetic data and demonstrate the method's robustness to medium and strong distribution shifts. For real-world applications, we recommend using the entire rigged transformations dataset.

For the occupancy network, we employed the MLP architecture described by (Amiranashvili et al., 2022), which features eight linear layers, each of 128 dimensions and dropout $p = 0.2$. The initial

training of the occupancy network was conducted on an anatomical atlas. This intensive training process required 48 hours on a single NVIDIA RTX A6000 GPU, equipped with 48 GB of memory. This pretrained network was then used to fine-tune both Dropsembles and MC dropout models. Additionally, we trained four distinct instances of Ensembles without dropout, with each instance requiring 48 hours of training. A key distinction between Dropsembles and Ensembles is that Dropsembles leverage a single pretrained model from dataset A, significantly reducing computational demand by a factor equal to the number of thinned networks. This results in a *substantial difference in total training time*, as the pre-training on dataset A is much more time-consuming than the fine-tuning on dataset B (see Table 2 and Appendix E). Additionally, this training time on dataset A remains constant for Dropsembles as the number of thinned networks increases, while it grows linearly for Ensembles.

As the target dataset B, we use a publicly available dataset of MR+CT images from 20 subjects (Cai et al., 2015). We employed the same segmentation network as (Turella et al., 2021) to obtain segmentations of 5 vertebrae, 5 discs, and the spinal canal (Pang et al., 2020). These automatically generated segmentations of high-quality MRI samples were used as the ground truth for the sparse 3D reconstruction task. To create sparse inputs, we removed the same set of sagittal slices as in the atlas dataset. We randomly selected 3 subjects for the consequent fine-tuning and testing.

The MR dataset described above lacks ground truth segmentations, which complicates numerical evaluation. The goal of the model is to impute missing or misclassified parts using anatomical atlas

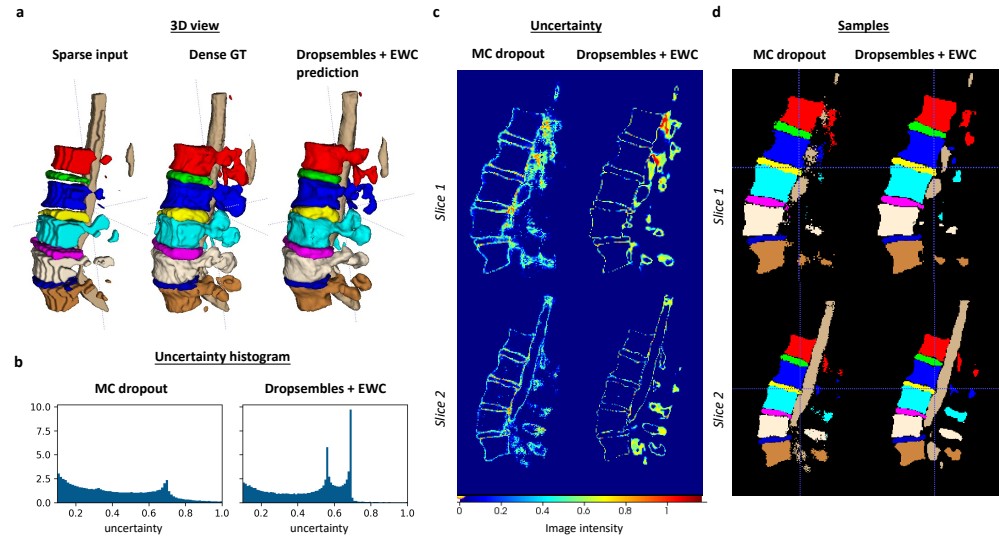

Figure 4: Lumbar spine reconstruction example on Subject 2. a) 3D-rendered views of sparse inputs (bicubic upsampling), dense ground truth (GT), and predictions by our method. b) Histograms of uncertainty (entropy) values, truncated between 0.1 and 1 for visibility. c) Examples of uncertainty estimates for two different sagittal slices of the 3D volume. d) For each method, network predictions are randomly sampled and the corresponding reconstructions are depicted.

priors, which, while enhancing reconstruction, are typically marked as incorrect in standard segmentation metrics. To facilitate a more accurate assessment of reconstruction metrics, we introduce an intermediate benchmark. For this purpose, we utilize three random rigged deformations from the atlas that were not exposed during training and apply a doubled level of erosions. This strong augmentation provides a challenging input for dataset B and allows to model a significant distribution shift aimed at rigorously testing the model under adverse conditions.

Numerical evaluation in Table 2 demonstrates consistent improvement of our method upon MC-dropout and comparable performance to Ensembles. In order to investigate the performance of the proposed method on MR dataset, we perform a detailed qualitative analysis in Figure 4. A consistent and distinct pattern not captured by the metrics alone, stands out from the qualitative assessment: MC-dropout tends to produce noisy reconstructions, as illustrated in Figure 4c-d, and generally

yields predictions that lack coherent, continuous shapes. This behaviour is partially captured in metric "DSC avg" in Table 2 - dice score evaluated on individual samples of the network instances. This observation is notable because, although models trained with conventional dropout generate reasonable predictions, their use in uncertainty estimation undermines the fundamental objective of 3D modeling. Samples drawn from the dropout distribution do not yield plausible shapes, as evidenced in Figure 4d. In contrast, our model, which draws from deep ensembles, does not exhibit these limitations. Furthermore, increasing the number of samples in MC-dropout does not resolve this issue but rather leads to higher computational demands, as documented in Table 2.

EWC demonstrates only minor improvements in terms of Dice score, while the difference in Hausdorff distance is more noticeable. We believe this is because Hausdorff distance better captures small discrepancies that the Dice score overlooks. Dice is less sensitive to small differences in reconstruction, as the majority of structures are well reconstructed by both methods. Qualitative inspection in Figure 5 confirms these subtle variations, which are crucial in medical applications where even small differences can significantly impact treatment planning.

Table 2: Comparison of methods fine-tuned on dataset B of lumbar spine experiment. Evaluations are performed on the corrupted atlas. Metrics are reported for dataset B. Baseline is a MC-dropout model trained on dataset A (w/o fine-tuning).

| Method | DSC [%] ↑ | DSC avg [%] ↑ | HD ↓ | ECE [%] ↓ | Train-A / Tune-B [h] |
|---|---|---|---|---|---|
| Baseline | $65.0 \pm 2.0$ | $63.9 \pm 1.8$ | $17.5 \pm 1.2$ | $25.2 \pm 0.4$ | 48 / - |
| MCdropout | $84.8 \pm 2.8$ | $83.0 \pm 2.0$ | $18.15 \pm 6.46$ | $\underline{3.9} \pm 2.4$ | 48 / 3.5 |
| Dropsembles | $\underline{86.8} \pm 2.4$ | $\underline{86.3} \pm 2.3$ | $11.4 \pm 1.8$ | $4.6 \pm 1.7$ | 48 / 10 |
| Dropsembles + EWC | $\mathbf{86.9} \pm 2.4$ | $\mathbf{86.4} \pm 2.3$ | $\underline{10.9} \pm 2.1$ | $4.5 \pm 1.7$ | 48 / 11 |
| Ensembles | $86.5 \pm 3.1$ | $86.0 \pm 2.9$ | $\mathbf{10.5} \pm 2.8$ | $\mathbf{3.9} \pm 2.2$ | 192 / 10 |

# 6 DISCUSSION AND CONCLUSION

**Strengths** In this study, we advanced sparse 3D shape reconstruction for high-precision applications by introducing uncertainty modeling. We developed a flexible framework that facilitates uncertainty-aware fine-tuning of models and showcased its utility in reconstructing the lumbar spine from sparse and corrupted MRI data. Our observations suggest that traditional uncertainty methods like MC dropout are not ideal for implicit shape reconstruction, as they tend to undermine the basic principles of implicit functions. However, our Dropsemble method effectively addresses these limitations, providing a promising alternative for uncertainty modeling. Additionally, Dropsembles are significantly more efficient than Ensembles during the resource-heavy pre-training phase, allowing for substantial reductions in resource usage without significant loss in performance.

**Limitations** During the fine-tuning stage, Dropsembles face the same computation demands as Ensemble models.

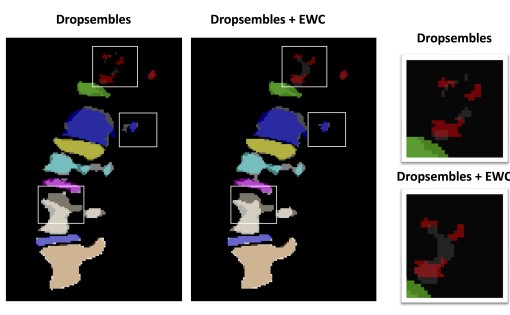

Figure 5: Example in the performance difference between Dropsembles w/ and w/o EWC. Grey segmentations are produced by Dropsembles w/ or w/o EWC regularization overlayed with colorful ground-truth segmentations. EWC better captures subtle details in modeling thin structures, such as vertebra processes.

**Future directions** In this work, we focus on the epistemic uncertainty of the decoder, assuming the encoder to be frozen. An interesting future direction is to combine Dropsembles with the uncertainty in the encoder to comprehensively cover all aspects of uncertainty modeling in sparse 3D shape reconstruction.

ACKNOWLEDGMENTS

This study was financially supported by: 1. Personalized Health and Related Technologies (PHRT), project number 222 and 643, ETH domain, 2. ETH AI Center.

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

## A    APPENDIX: ELASTIC WEIGHT CONSOLIDATION

Elastic Weight Consolidation (EWC) was introduced to address catastrophic forgetting in continual learning (Kirkpatrick et al., 2016). EWC is a regularization that protects crucial parameters in a network when learning a new task, in order to avoid catastrophic forgetting.

Assume two distinct tasks, $A$ and $B$, with their respective datasets $\mathcal{D}_A$ and $\mathcal{D}_B$, where $\mathcal{D}_A \cap \mathcal{D}_B = \emptyset$. The tasks are learned sequentially without access to previous task datasets. First, task $A$ is learned by training a neural network on $\mathcal{D}_A$, resulting in a set of optimal weights $\hat{\theta}_A$. When learning task $B$ using dataset $\mathcal{D}_B$, EWC regularizes the weights so they remain within a region in the parameter space that led to good accuracy for task $A$. The justification of the method is based on probabilistic principles. Given a combined dataset $\mathcal{D} := \mathcal{D}_A \cup \mathcal{D}_B$, applying Bayes' rule yields:

$$\log p(\theta|\mathcal{D}) = \log p(\mathcal{D}|\theta) + \log p(\theta) - \log p(\mathcal{D}) \tag{3}$$

where $p(\mathcal{D}|\theta)$ corresponds to the likelihood over the entire dataset and $p(\theta)$ is the user-defined prior over the weights of the network. Rearranging equation (3) yields

$$\log p(\theta|\mathcal{D}) = \log p(\mathcal{D}_B|\theta) + \log p(\theta|\mathcal{D}_A) - \log p(\mathcal{D}_B) \tag{4}$$

where it can be observed that all information from $\mathcal{D}_A$ is contained in the posterior $p(\theta|\mathcal{D}_A)$, which is usually intractable. Therefore, the EWC method employs the Laplace approximation to the posterior, a process conducted during the training of task $A$. The resulting approximated posterior is modeled as a Gaussian distribution, with its mean represented by $\hat{\theta}_A$ and covariance matrix $\Sigma_A = (F(\hat{\theta}_A) \circ I)^{-1}$ where $F(\hat{\theta}_A)$ denotes the Fisher information matrix evaluated over data set $\mathcal{D}_A$ at $\hat{\theta}_A$.

Next, in stage $B$, the optimization of parameter $\theta$ incorporates the approximation of the posterior obtained during task $A$ as a constraint in the optimization process. Following the equation (4), this approximation, $\log p(\theta|\mathcal{D}_A)$, serves as a regularizer in the learning objective:

$$\hat{\theta}_B = \arg\min_{\theta} \hat{R}_{\mathcal{D}_B}(\theta) + \lambda(\theta - \hat{\theta}_A)^T (F(\hat{\theta}_A) \circ I)(\theta - \hat{\theta}_A) \tag{5}$$

where $\hat{R}(\theta)$ corresponds to the likelihood term $\log p(\mathcal{D}_B|\theta)$, $\lambda$ is a hyperparameter, $I$ represents the identity matrix, and $\circ$ describes the Hadamard product.

## B    APPENDIX: METRICS

Uncertainty estimates for regression tasks involve using unbiased estimates of the modes of the approximated posterior predictive distribution. In neural networks, this corresponds to performing multiple stochastic forward passes. For classification tasks, three main methods for uncertainty estimates are variational ratios, predictive entropy, and mutual information (Gal, 2016). In our experiments, we opt for predictive entropy.

**Dice score**    The Dice score, also known as the Dice Similarity Coefficient (DSC), measures the similarity between two data sets, commonly used in medical imaging to evaluate segmentation accuracy. Defined as DSC $= \frac{2 \times |X \cap Y|}{|X| + |Y|}$, where $X$ and $Y$ represent the ground truth and predicted segmentation sets, respectively. The score ranges from 0 to 1, with 1 indicating perfect agreement and 0 representing no overlap.

**Reliability diagrams and Expected Calibration Error**    Reliability diagrams are graphical tools used in uncertainty modeling to assess the calibration of probabilistic predictions. They plot predicted probabilities against empirical frequencies, allowing for visual inspection of how well the predicted probabilities of a model correspond to the actual outcomes. A perfectly calibrated model would align closely with the diagonal line from the bottom left to the top right of the plot, indicating that the predicted probabilities match the observed probabilities. Reliability diagrams are computed by binning predicted probabilities into intervals. For each bin, the mean predicted probability is plotted against the observed frequency of the corresponding outcomes. This involves calculating the

proportion of positive outcomes in each bin and plotting these against the average predicted probability for the bin. The closer the points lie to the diagonal line from the bottom left to the top right, the more calibrated the model is considered. Reliability diagrams are closely related to the Expected Calibration Error (ECE), which quantitatively assesses a model's calibration. ECE is computed as a weighted average of the absolute differences between the predicted probabilities and the actual outcome frequencies across different bins used in reliability diagrams. Each bin's weight corresponds to the number of samples it contains. Thus, while reliability diagrams provide a visual interpretation of model calibration, ECE offers a single numerical value summarizing the calibration error across all bins.

## C  APPENDIX: TRAINING DETAILS

**Toy experiment**   We generated two-dimensional datasets A and B with a sinusoidal decision boundary and Gaussian noise. A moderate distribution shift was modeled between the datasets by adjusting the support values of $x_1$: $x_1 \in [-0.75, 0.7]$ for dataset A and $x_1 \in [-0.5, 2.0]$ for dataset B. We created 1000 training samples and 500 test samples for dataset A, and 50 train and 500 test samples for dataset B.

We used a consistent model architecture across experiments—a straightforward 3-layer MLP with 256 hidden units in each layer. For methods using dropout, a dropout layer ($p = 0.3$) followed each linear layer. For ensemble methods, 4 separate networks were trained on dataset A. For ensembles and Dropsembles we used 4 network instances for fine-tuning and inference. For MC dropout we used 100 samples at inference. We trained for 800 epochs for training with a learning rate $1e - 3$ and 600 epochs for tuning with a learning rate $5e - 3$.

**MNIST experiment**   First, we trained a small autoencoder, consisting of three convolutional layers in the encoder, only on dataset A. The encoder does not have dropout layers. The encoder was trained with cross-entropy loss for 50 epochs with a learning rate $0.01$ and a cosine warmup scheduler. After this initial training, the encoder was kept fixed (frozen) for all subsequent experiments, and the decoder was discarded. This encoder now serves to generate a latent representation of the input data, which is then supplied to an occupancy network.

The convolutional encoder was trained on Dataset A and consequently frozen for all the rest of the experiments. Pooling layer was applied to produce same latent code for all 2D coordinates in a shape.

The 8-layer MLP occupancy network was trained with cross-entropy loss for 50 epochs on dataset A with a learning rate $0.005$ and a cosine warmup scheduler. For ensemble methods, 4 separate networks were trained on dataset A. For fine-tuning on dataset B we used same learning rate but tuned the networks for 30 epochs only. For ensembles and Dropsembles we used 4 network instances for fine-tuning and inference.

**ShapeNet experiment**   For Dataset A, we selected 1,000 random samples from the ShapeNet dataset's airplane category. For Dataset B, we randomly selected 10 unseen shapes and applied random erosion to simulate real-world distribution shifts. Examples of eroded and high-quality target shapes are shown in Figure 12. For both datasets, occupancies at a resolution of 128 were provided to the encoder. SDF values for Dataset A were estimated from ground-truth high-quality meshes. For Dataset B, meshes were reconstructed from the eroded shapes and subsequently converted into SDF representations.

The convolutional encoder was trained on Dataset A and consequently frozen for all the rest of the experiments. Bi-linear upsampling was applied to query latent vectors for specific 3D coordinates as proposed in convolutional occupancy networks architecture.

The 8-layer DeepSDF MLP network was trained on Dataset A for 100 epochs using a clipped $L1$ loss (with clipping parameter $\delta = 0.1$), a learning rate of $0.001$, and a step scheduler as proposed in (Park et al., 2019a). When applicable, dropout was added after each activation layer, as suggested in DeepSDF. For ensemble methods, four separate networks were trained on Dataset A. Fine-tuning on Dataset B was performed using a non-clipped $L1$ loss with a slower learning rate of $0.0001$ and limited to 75 epochs. For ensembles and Dropsembles, four network instances were employed

during fine-tuning and inference. In the SIREN experiment, ReLU activations were replaced with periodic activation functions (Sitzmann et al., 2020), with dropout applied only after the 4th and 6th layers in the decoder network.

**Lumbar spine experiment** All experiments in this section were performed on NVIDIA RTX A6000 GPU, equipped with 48 GB of memory. The networks were trained at 16-mixed precision due to memory constraints.

First ReconNet encoder was trained with cross-entropy loss on a full training split of rigged atlas dataset to obtain a "frozen" encoder. We used a learning rate 0.01 and trained for 100 epochs with early stopping. Bi-linear upsampling was applied to query latent vectors for specific 3D coordinates as proposed in convolutional occupancy networks architecture.

The occupancy network architecture incorporates skip connections and ReLU activations, with dropout layers ($p = 0.2$) following each linear layer except the last. To effectively model the entire lumbar spine, we adapted the strategy from (Peng et al., 2020) by replacing the learnable latent vector with an output from a pretrained convolutional encoder. Specifically, we performed bilinear upsampling on the output of this frozen encoder to generate a detailed latent representation for each voxel. This representation, coupled with 3-dimensional voxel coordinates, was provided as input to the MLP. We used cross-entropy loss for training and fine-tuning the occupancy network. The network was trained for 100 epochs on dataset A with early stopping applied after 68 epochs. Learning rate 0.001 and batch size 32 were used for training, where each batch we used only [64, 64, 64] random voxels. For fine-tuning on dataset B we used the same of parameters for all the methods: learning rate 0.001 and tuned it for 50 epochs without early stopping. Results evaluated at the checkpoint of the last epoch are presented throughout the paper.

# D APPENDIX: SELECTING OPTIMAL REGULARIZATION STRENGTH

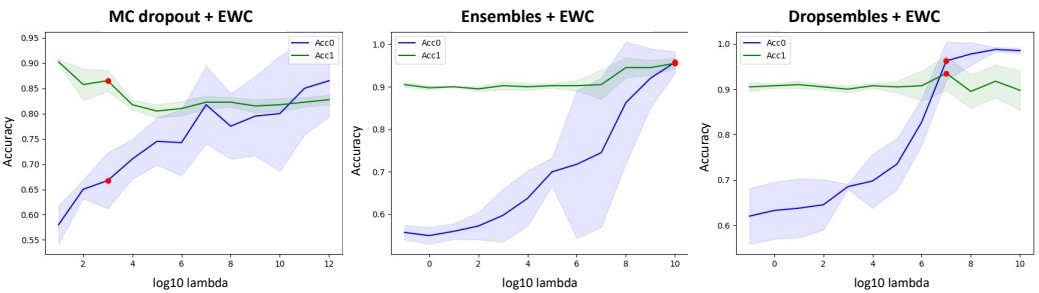

Figure 6: Toy classification example ablation for EWC regularization strength. Averages and confidence over 3 random seeds. Red dot represents the optimal lambda selected for the main figure.

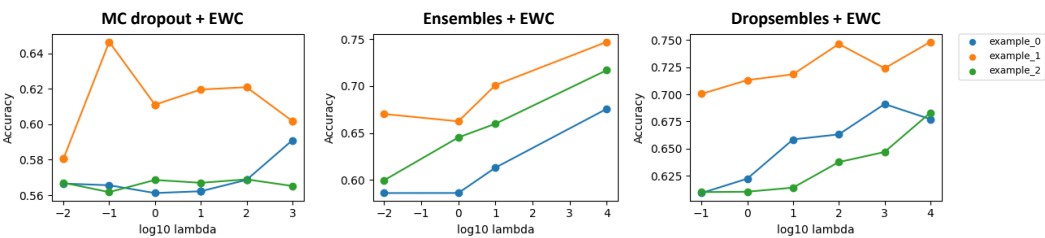

Figure 7: Corrupted MNIST reconstruction example ablation for EWC regularization strength. Dots correspond to different samples.

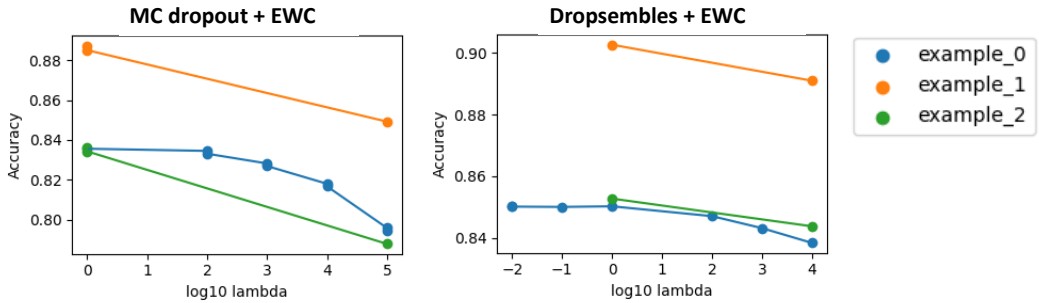

Figure 8: Corrupted atlas reconstruction example ablation for EWC regularization strength. Dots correspond to different samples. Due to high computational demand we were not able to evaluate all examples on the full grid.

# E APPENDIX: ADDITIONAL EXPERIMENTAL RESULTS

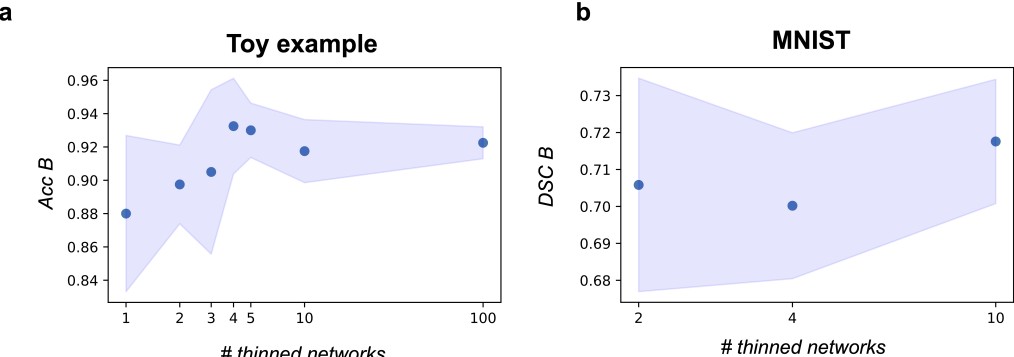

Figure 9: Example in the performance difference between Dropsembles w/ and w/o EWC: tiny details in patient-specific variability are not captured well in DSC metrics, but are apparent on the qualitative examples and Hausdorff distance (HD). Grey segmentations are produced by Dropsembles w/ or w/o EWC regularization overlayed with colorful segmentations of the HQ ground-truth.

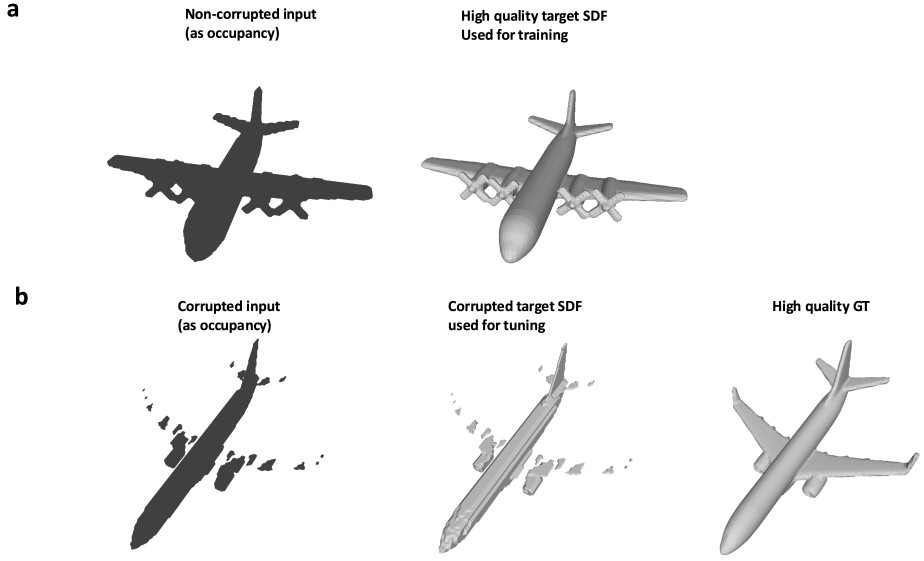

Figure 10: Examples of ShapeNet samples. a) Dataset A; b) Dataset B.

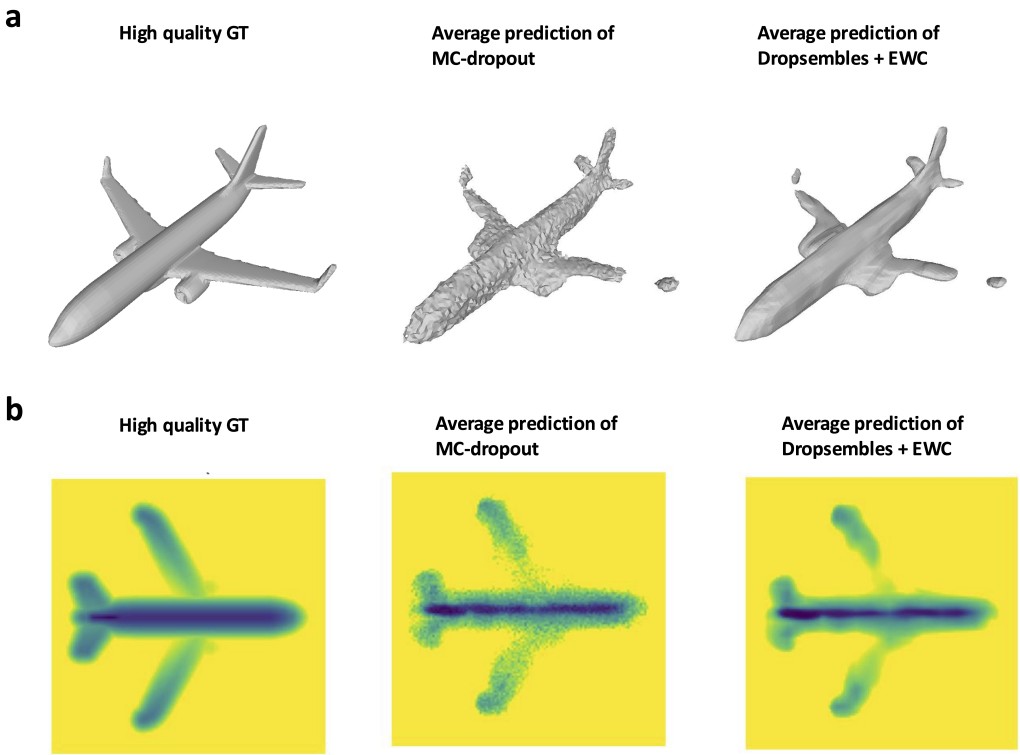

Figure 11: Comparison of average predictions b/w MC Dropout and Dropsembles. Dropout predictions even after averaging are not smooth. a) Meshes produced by averaged predictions; b) A slice of SDF representation produced by averaging predicted samples.

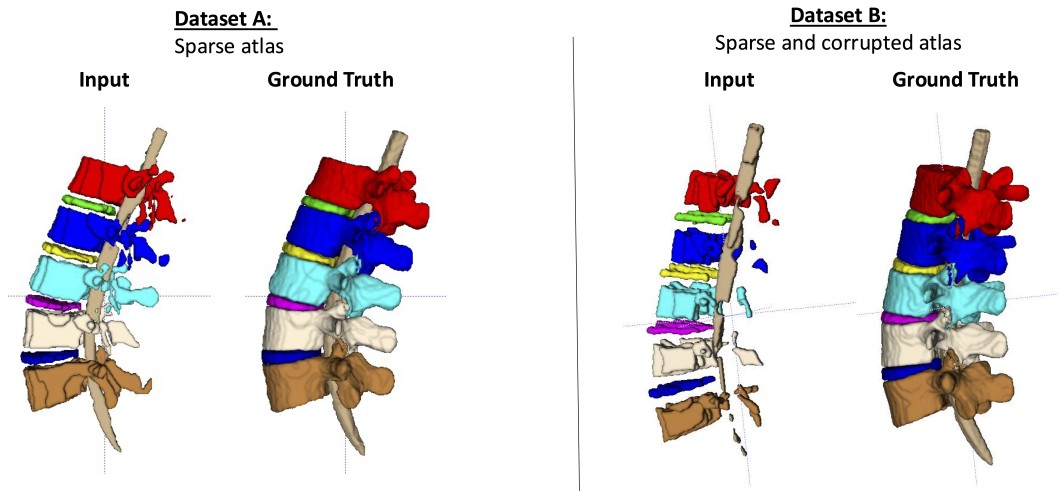

Figure 12: Examples of Corrupted Atlas dataset.

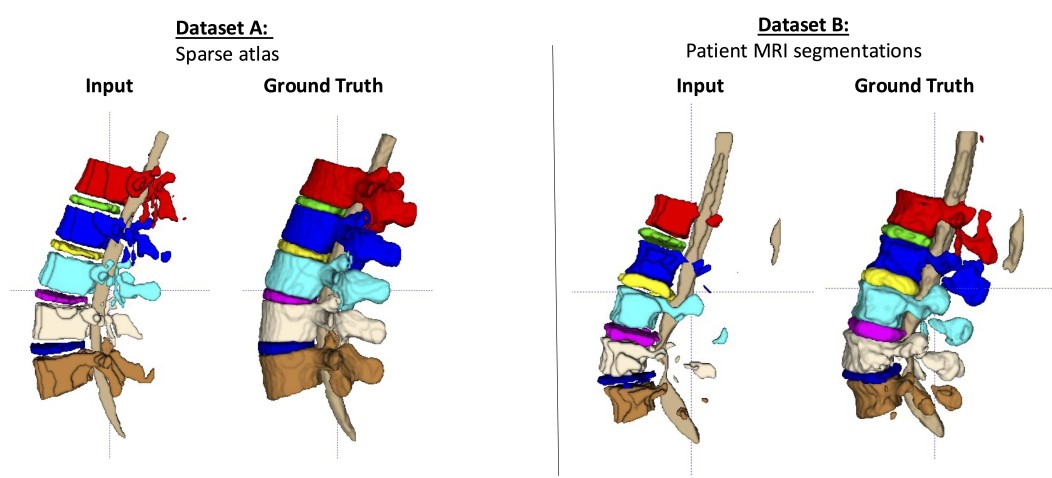

Figure 13: Example of Atlas to MRI segmentations dataset.

Table 3: ShapeNet: Results for models trained on dataset A only. All metrics are reported in percentages [%].

| Method | mDICE [%] ↑ | mIoU [%] ↑ | ECE [%] ↓ |
|---|---|---|---|
| Dropout | 78.4 ± 2.4 | 64.5 ± 3.2 | **31.66** ± 3.37 |
| Dropout - SIREN | **78.5** ± 3.3 | **64.8** ± 4.4 | 31.94 ± 3.89 |
| Ensembles | 76.4 ± 2.9 | 61.9 ± 3.7 | 36.33 ± 3.11 |
| Ensembles - SIREN | 75.9 ± 3.7 | 61.3 ± 4.8 | 36.44 ± 3.85 |

Table 4: ShapeNet with SIREN activations: Results for models trained on dataset A and B (fine-tuned on dataset B). All metrics are reported in percentages [%].

| Method | mDICE [%] ↑ | mIoU [%] ↑ | ECE [%] ↓ |
|---|---|---|---|
| Dropout | 80.2 ± 2.4 | 67.0 ± 3.3 | 27.99 ± 2.85 |
| Dropsembles | 83.8 ± 2.2 | 72.2 ± 3.2 | 21.04 ± 3.04 |
| Ensembles | 83.4 ± 2.5 | 71.6 ± 3.6 | 24.44 ± 3.03 |
| Dropout + EWC | 83.3 ± 2.2 | 71.5 ± 3.2 | 19.27 ± 2.74 |
| Dropsembles + EWC | **84.5** ± 2.3 | **73.2** ± 3.4 | **17.73** ± 3.05 |
| Ensembles + EWC | 80.9 ± 2.5 | 68.0 ± 3.5 | 28.95 ± 3.07 |

Table 5: Detailed comparison of training time on lumbar spine experiment. We report the results in GPU-hours [h].

| Method | Training on atlas [h] | Tuning on corrupted atlas / lumbar spine MRI [h] | Total [h] |
|---|---|---|---|
| Baseline | 48 | - / - | 48 / 48 |
| MCdropout | 48 | 3.5 / 22 | 51.5 / 99.5 |
| Dropsembles | 48 | 2.5 / 21 (per network) x4 | 58 / 132 |
| Dropsembles + EWC | 48 | 2.75 / 22 (per network) x4 | 58 / 136 |
| Ensembles | 48 (per network) x4 | 2.5 / 21 (per network) x4 | 202 / 276 |

Table 6: Comparison of methods tuned on dataset B of lumbar spine MR dataset. Dice Score Coefficient (DSC), Hausdorff distance (HD), Dice Score Coefficient average per network sample (DSC avg), Expected Calibration Error (ECE), and total inference time (time) are reported for dataset B. Baseline is MC dropout trained on dataset A.

| | Method | DSC [%] ↑ | DSC avg [%] ↑ | HD ↓ | ECE [%] ↓ | time [sec] ↓ |
|---|---|---|---|---|---|---|
| Subject 1 | Baseline | 75.8 | 74.0 ± 0.0 | 18.2 | 4.7 | 161 |
| | MCdropout | 93.5 | 92.3 ± 0.0 | 6.3 | **3.6** | 159 |
| | Dropsembles | **93.9** | **93.5** ± 0.1 | 6.2 | 4.4 | 20 |
| | Dropsembles + EWC | **93.9** | **93.5** ± 0.0 | **6.0** | 4.2 | 21 |
| | Ensembles | **93.9** | **93.5** ± 0.2 | 24.3 | 4.3 | 21 |
| Subject 2 | Baseline | 35.9 | 36.4 ± 0.1 | 34.2 | 33.6 | 30 |
| | MCdropout | 91.3 | 89.7 ± 0.0 | 15.2 | **4.9** | 160 |
| | Dropsembles | **92.0** | **91.5** ± 0.1 | **9.4** | 6.1 | 19 |
| | Dropsembles + EWC | **92.0** | **91.5** ± 0.1 | 9.5 | 6.1 | 19 |
| | Ensembles | 91.9 | 91.4 ± 0.1 | 25.2 | 6.0 | 21 |
| Subject 3 | Baseline | 76.5 | 74.2 ± 0.0 | 19.5 | **3.7** | 159 |
| | MCdropout | 92.5 | 91.3 ± 0.0 | 7.1 | 4.9 | 158 |
| | Dropsembles | 92.6 | 92.2 ± 0.0 | 8.2 | 5.3 | 19 |
| | Dropsembles + EWC | 92.6 | 92.2 ± 0.0 | 6.7 | 5.2 | 19 |
| | Ensembles | **92.8** | **92.5** ± 0.1 | **6.5** | 5.6 | 21 |

Table 7: Comparison of methods tuned on dataset B of lumbar spine experiment. Evaluations are performed on the corrupted atlas. Dice Score Coefficient (DSC), Hausdorff distance (HD), Dice Score Coefficient average per network sample (DSC avg), Expected Calibration Error (ECE), and total inference time (time) are reported for dataset B. Baseline is MC dropout trained on dataset A.

| | Method | DSC [%] ↑ | DSC avg [%] ↑ | HD ↓ | ECE [%] ↓ | time [sec] ↓ |
|---|---|---|---|---|---|---|
| **Corrupted Atlas 1** | Baseline | 65.5 | $64.4 \pm 0.0$ | 16.8 | 24.7 | 172 |
| | MCdropout | 83.5 | $81.7 \pm 0.0$ | 14.3 | **4.6** | 160 |
| | Dropsembles | **85.0** | **84.7** $\pm 0.2$ | **13.2** | 5.7 | 21 |
| | Dropsembles + EWC | **85.0** | **84.7** $\pm 0.2$ | 13.3 | 5.6 | 21 |
| | Ensembles | 84.5 | 84.2 $\pm 0.7$ | 13.4 | 5.0 | 21 |
| **Corrupted Atlas 2** | Baseline | 67.2 | $65.8 \pm 0.0$ | 16.5 | 25.6 | 164 |
| | MCdropout | 88.6 | $85.7 \pm 0.0$ | 9.4 | **0.8** | 170 |
| | Dropsembles | 90.2 | **89.6** $\pm 0.0$ | 8.9 | 2.3 | 22 |
| | Dropsembles + EWC | **90.3** | **89.6** $\pm 0.0$ | **8.1** | 2.1 | 22 |
| | Ensembles | 90.0 | 89.3 $\pm 0.7$ | 7.8 | 1.3 | 22 |
| **Corrupted Atlas 3** | Baseline | 62.3 | $61.4 \pm 0.0$ | 23.5 | 25.4 | 163 |
| | MCdropout | 83.7 | $81.5 \pm 0.0$ | 15.4 | **4.8** | 162 |
| | Dropsembles | 85.2 | **84.8** $\pm 0.1$ | 12.0 | 5.9 | 22 |
| | Dropsembles + EWC | **85.3** | **84.8** $\pm 0.2$ | 11.2 | 5.8 | 22 |
| | Ensembles | 84.9 | 84.5 $\pm 0.7$ | **10.2** | 5.3 | 22 |

