# OpenReview forum: "Uncertainty modeling for fine-tuned implicit functions"
_ICLR.cc/2025/Conference — ICLR 2025 Poster_

### Official Review · Reviewer_Yvca · 2024-10-28

**Soundness:** 2
**Presentation:** 3
**Contribution:** 2
**Rating:** 3
**Confidence:** 4

**Summary:**

In this work, authors introduce Dropsembles, a novel method for uncertainty estimation in tuned implicit functions. They demonstrate the efficacy of this approach through a series of experiments, starting with toy examples and progressing to a real-world scenario. Our results show that Dropsembles achieve the accuracy and calibration levels of deep ensembles but with significantly less computational cost.

**Strengths:**

1.  The topic of this work is really interesting. By utilizing an implicit shape model, points with sparse shapes can be densified with refined shape representations.
2.  This is the first work to model epistemic uncertainty in the implicit decoder during finetuning from synthetic data.

**Weaknesses:**

1.  The core concept of this work is uncertainty modeling. There are two innovative designs proposed in this paper. For the Elastic weight consolidation strategy, it seems that no remarkable improvements are done. Authors simply employ EWC to adapt pretrained models from dataset A. The other component is Dropsembles, please provide a more detailed description on its contribution.
2.  Since Dropsembles is aimed for the tradeoff between computational costs and accuracy. However, no quantitative results are given in the experimental part. Authors should carefully list out the comparison between dropsembles and other methods on uncertainty estimation.
3.  The improvement on lumbar spine seems to be marginal, maybe a detailed analysis is required.
4.  The effectiveness of this method is highly required for the evaluation on more challenging datasets, such as Medshapenet, etc.

**Questions:**

Please refer to the weakness part.

---

> ### Author Response · Authors · 2024-11-15
> **Response to Reviewer Comment on Dropsembles Description**
>
> We appreciate the reviewer's request for more detailed information about Dropsembles. We would like to point out that comprehensive information about Dropsembles is already provided throughout our paper in several key sections:
>
> **1. Initial Overview.** As stated in our contributions (line 086), we introduce Dropsembles as a novel method for uncertainty estimation in fine-tuned implicit functions.
>
> **2. Visual Representation.** Figure 1 provides a clear overview of the method's architecture and workflow, illustrating how Dropsembles operates in the context of training with a dense prior and fine-tuning on a sparse dataset.
>
> **3. Formal Definition and Mathematical Framework.** The formal definition of "Dropsembles" is given in Section 4.1 Algorithm 1, where all the components of the algorithm are well defined through the mathematical framework that we introduce along Section 3.1 to 3.3 for the optimization problem, as well as the underlying mathematical modeling of our problem at the start of section 4 (line 232-251).
>
> **4. Theoretical Justification.** The theoretical justification for our method is explained in section 4.1 "Dropsembles", where we detail how our approach combines the advantages of both dropout and deep ensembles while addressing their respective limitations. Specifically, we explain how Dropsembles:
>
> - Moderates the computational demands typically associated with ensembles while maintaining prediction accuracy (lines 273-275)
> - Creates an ensemble of "thinned" networks that share weights, effectively training a collection of correlated networks simultaneously (lines 273-276)
> - Treats the ensemble as a uniformly weighted mixture model for inference and uncertainty estimations (lines 276-278)
> - Provides comparable prediction performance metrics and expected calibration error to traditional ensembles, despite the relaxation of independence between network instances (lines 278-279)
>
> **5. Relationship to Existing Literature.** As discussed in our Related Work section (lines 067-080), we position our work in the context of existing uncertainty estimation methods. While MC dropout (Gal and Ghahramani, 2016) and deep ensembles (Lakshminarayanan et al., 2017) are commonly used baselines, they each have limitations. MC dropout often underestimates uncertainty and requires multiple forward passes, while deep ensembles are computationally intensive. Dropsembles bridges this gap by combining the efficiency of dropout with the robust uncertainty estimates of ensembles, while being particularly suited for fine-tuning scenarios where computational resources are constrained.
>
>
> We believe these sections together provide a thorough and well-structured description of our method. However, if the reviewer has specific aspects of Dropsembles they would like us to elaborate on, we would be happy to provide additional details.

---

> ### Author Response · Authors · 2024-11-15
> **Response to Request for Computational Cost and Accuracy Comparisons**
>
> We kindly ask the reviewer to revisit our manuscript, because all of the analysis they mention, are already included in original submission. Below, we list pointers to the exact location where the reviewer can find quantitative comparison as well as a discussion for the requested results:
>
> ## Computational Costs (Table 2, lines 472-490)
> Training time comparisons for the lumbar spine experiment clearly show the computational advantage of Dropsembles:
> - Ensembles: 192 hours for training on dataset A
> - Dropsembles: 48 hours for training on dataset A
> - Both methods require similar fine-tuning time on dataset B (approximately 10 hours)
>
> ## Accuracy and Uncertainty Metrics
> We provide comprehensive comparisons across multiple experiments:
>
> 1. **Toy Classification** (Table 1):
> - Accuracy and ECE metrics for MC dropout, Dropsembles, and Ensembles
> - Dropsembles achieves comparable performance to Ensembles (90.5% vs 90.0% accuracy)
>
> 2. **MNIST Reconstruction** (Table 1):
> - DSC and ECE metrics for all methods
> - Dropsembles (64.1% DSC, 6.1% ECE) performs similarly to Ensembles (62.6% DSC, 5.8% ECE)
>
> 3. **Lumbar Spine** (Table 2):
> - DSC, Hausdorff distance, and ECE metrics
> - Dropsembles achieves comparable performance to Ensembles while requiring only 25% of the training time
>
> ## Additional Detailed Results (Appendix)
> For a more comprehensive analysis, we provide extensive additional experimental results in Appendix:
> - Table 3: Detailed training time comparisons for all methods
> - Table 4: Per-subject performance metrics on lumbar spine MR dataset
> - Table 5: Extended evaluation metrics on corrupted atlas data
>
> These results demonstrate that Dropsembles achieves the desired balance between computational efficiency and accuracy in uncertainty estimation.

---

> ### Author Response · Authors · 2024-11-15
> **Response to Request on additional analysis on Lumbar spine**
>
> We have a detailed discussion on the “marginal improvement” on the Lumbar spine dataset in [lines 472-484], where we discuss the role of different metrics (e.g. Dice score vs Hausdorff distance) and support our hypothesis with detailed qualitative evaluations [Figure 4]. We would kindly refer the reviewer to this figure and the discussion and we would be happy to provide additional details to address further concerns.

---

> ### Author Response · Authors · 2024-11-26
> **Response to Reviewer Comment on Evaluation on more challenging datasets**
>
> We appreciate your suggestion and would like to refer to the "General Response" and the revised version of the manuscript, where we added a comparison on the ShapeNet dataset.

---

### Official Review · Reviewer_BMvL · 2024-11-03

**Soundness:** 3
**Presentation:** 4
**Contribution:** 2
**Rating:** 8
**Confidence:** 4

**Summary:**

This paper explores the topic of uncertainty modeling for decoder fine-tuning in implicit neural representations for occupancy representations. The presented method employs a combination of widely used uncertainty modeling concepts (i.e., dropout, ensembling) and the elastic weight consolidation concept stemming from continual learning, with the notion of fine-tuning, allowing to obtain multiple (fine-tuned) models at less training cost, with partially improved or comparable performance in modeling the uncertainty. This is demonstrated in experiments with toy examples (Sinusoidal decision boundary, MNIST) to medical shapes with the perseverance of reconstruction accuracy.

**Strengths:**

Originality: While the paper builds upon technical concepts from the uncertainty modeling community and does not introduce novel technical components, I believe the explored application and conclusions are novel and impactful.

Quality: The paper is very well structured and intuitive. The presentation and writing of the method are very clear, and in combination with the experiments, the method is sound.

Clarity: The paper is well written in all sections, especially in related work and methods, making it a charm to read. It really flows. I also really enjoyed how well the method was put into the context of related work (i.e. very thoroughly research in different areas) and how the method was described in a simple and comprehensive way. Moreover, the authors detail every aspect of the experimental design in the main paper, and together with the supplementary this makes the evaluation of the presented method easy to follow. This, together with the accessibility of the source code, facilitates reproducability of the experiments.

Significance: While implicit neural representations are often trained - individually - for (single) images and other signals, cohort-based training is quite common for shape completion in SDFs and occupancy representations. Given the prevalence of encoder-/decoder in these settings, this paper constitutes a meaningful exploratory study in the modeling of decoder uncertainty.

**Weaknesses:**

1) Related Work:

As detailed above, the paper is very well written and conducts a very thorough related work section. However, I think it would be important to touch upon one more area - i.e. the more recent architectures used in cohort-based training that also use encoder frameworks. Specifically, it would be interesting to discuss the advancements in [1,2], given that these constitute more recent alternatives to the DeepSDF architecture. As both works [1,2] use encoder-/decoder settings in combination with weight modulation approaches, this would also add emphasis to the relevance of this work.

2) Experimental Design: While the experimental design is sound, I believe the paper could have explored a higher variation of datasets and application scenarios.

(a)  Experiments not only for Re-LU-based networks but also for other activation functions:
While ReLU networks were routinely used in [3,4,5] medical applications, the INR community has shifted towards other activation functions [6] or embedding/projection functions [7,8] (particularly for images). While not all (novel) shape modeling applications use these (yet?), I believe it would have been interesting to explore if the same holds true for, e.g., SIREN [6]. This ablation would have been simple, as it merely requires a change in the MLP and DeepSDF [3], which also works well with sinusoid activation functions [6]. Moreover, it has been comparatively less explored if dropout and other anomaly detection frameworks are useful in this context. Thus, for decoder-based uncertainty modeling, for example, in the context of [2], such an ablation study would be ultimately interesting and relevant.

(b) Experiments on more common (shape) datasets, especially with SDFs: The authors state that their method is relevant for both SDF and occupancy representations but only conducts experiments for occupancy networks. Given that the training setup would not change much, I believe it would have been interesting to show experiments in an established shape dataset as well (e.g. (Med-) ShapeNet).

References:

[1] Dupont E, Loya H, Alizadeh M, Goliński A, Teh YW, Doucet A. Coin++: Neural compression across modalities. arXiv preprint arXiv:2201.12904. 2022 Jan 30.

[2] Mehta I, Gharbi M, Barnes C, Shechtman E, Ramamoorthi R, Chandraker M. Modulated periodic activations for generalizable local functional representations. InProceedings of the IEEE/CVF International Conference on Computer Vision 2021 (pp. 14214-14223).

[3] Park JJ, Florence P, Straub J, Newcombe R, Lovegrove S. Deepsdf: Learning continuous signed distance functions for shape representation. InProceedings of the IEEE/CVF conference on computer vision and pattern recognition 2019 (pp. 165-174).

[4] Mescheder L, Oechsle M, Niemeyer M, Nowozin S, Geiger A. Occupancy networks: Learning 3d reconstruction in function space. InProceedings of the IEEE/CVF conference on computer vision and pattern recognition 2019 (pp. 4460-4470).

[5] Amiranashvili T, Lüdke D, Li HB, Menze B, Zachow S. Learning shape reconstruction from sparse measurements with neural implicit functions. InInternational Conference on Medical Imaging with Deep Learning 2022 Dec 4 (pp. 22-34). PMLR.

[6] Sitzmann V, Martel J, Bergman A, Lindell D, Wetzstein G. Implicit neural representations with periodic activation functions. Advances in neural information processing systems. 2020;33:7462-73.

[7] Tancik M, Srinivasan P, Mildenhall B, Fridovich-Keil S, Raghavan N, Singhal U, Ramamoorthi R, Barron J, Ng R. Fourier features let networks learn high frequency functions in low dimensional domains. Advances in neural information processing systems. 2020;33:7537-47.

[8] Müller T, Evans A, Schied C, Keller A. Instant neural graphics primitives with a multiresolution hash encoding. ACM transactions on graphics (TOG). 2022 Jul 22;41(4):1-5.

**Questions:**

Importance of encoder design / Application in auto-decoder setting:

Q1: Given the suitability of INRs for non-grid data, it is sometimes challenging to design an encoder for non-grid data. How was the encoder used/designed for the MNIST case, where some pixels are occluded? Did the encoder see the entire image (e.g., with a CNN-based encoder)? Sorry if I missed this; I would really appreciate an answer, even if it does not change the comparability of the different methods/baselines.

Q2: In the INR community, auto-decoder settings are quite common as well (see DeepSDF or Mehta et al.). Could the same method also be used in this case, given that the encoding of, e.g., the shape or instance stems from the same MLP, or are there any theoretical limitations that prohibit a meaningful uncertainty in this case?

---

> ### Author Response · Authors · 2024-11-27
> **General Response Reviewer BMvL**
>
> We sincerely thank the reviewer for their thorough review and for providing constructive suggestions regarding the related work and experimental design. These insights have been invaluable in improving the manuscript.
>
> In response, we conducted the suggested experiments with DeepSDF on the ShapeNet dataset and further extended them with a SIREN ablation study (please refer to the "General Response" and the revised manuscript). Unfortunately, due to the page limit in the main text, we could not provide an elaborated discussion of methods [1] and [2], which we agree would have been an interesting addition. However, we have ensured that these papers are appropriately cited in the revised version.
>
> ---------
>
> ## Response to Importance of encoder design / Application in auto-decoder setting
>
> **Q1: How was the encoder used/designed for the MNIST case, where some pixels are occluded?**
>
> We agree with the reviewer that this is a valuable question, as multiple options are indeed possible. In our approach, we chose a straightforward strategy by setting occluded pixels to zero and using a convolutional neural network as the encoder. While this approach exposes the encoder to a distribution shift, it does not affect our conclusions since we focus on modeling uncertainty in the decoder. Another alternative could involve modeling the encoder with graph neural networks, which could better handle the structural relationships in the input data. However, since the encoder is frozen for all the methods, it doesn't affect the performance of the benchmarks.
>
> **Q2: Could the same method also be used auto-decoder case?**
>
> This is an important question that touches on a fundamental distinction between auto-decoder and encoder-decoder architectures in terms of uncertainty quantification.
>
> In auto-decoder settings, the optimization objective inherently couples the latent representation with the network parameters. Specifically, it produces a maximum a posteriori estimate for the latent embedding given the observed shape in Dataset B. This step completely eliminates the fine-tuning stage and thus represents a fundamentally different approach. In this case, individual network instances could be used to estimate uncertainty over the latent representation. However, the reliability of such uncertainties for a network trained solely on Dataset A and subjected to a strong distribution shift in Dataset B remains unclear and has not been demonstrated in previous work.
>
> In our approach, uncertainty is isolated specifically to the decoder's parameter space and is not directly dependent on the latent representation, making it clearer what the uncertainty represents. Additionally, fine-tuning is an essential part of our method, as it allows the network to leverage observed partial information to adjust its weights, thereby mitigating challenges posed by strong distribution shifts.
>
> In summary, while uncertainty quantification in auto-decoder settings is an interesting direction for future research, it would require a fundamentally different approach from ours, as the decoupling of encoder and decoder is central to both the functionality and interpretability of our method.
>
> ------------
> We hope these revisions demonstrate that our work is ready for acceptance. Should there be any further questions, we are more than happy to address them.

---

> ### Comment · Reviewer_BMvL · 2024-12-01
>
> Thank you for providing such a thorough rebuttal and for including the additional experiments on ShapeNet and SIREN. These additions significantly strengthen the paper, and I now feel that the claims are well-supported by the experiments. While the approach is relatively simple, I believe that simplicity is one of its strengths. While it targets (only) a specific area of uncertainty modeling, I do feel it's meaningful given that there is very little work in the context of INRs. This makes it a valuable contribution that I believe can lead to meaningful discussion at ICLR. I have decided to raise my score (from 6 to 8).

---

### Official Review · Reviewer_iePp · 2024-11-03

**Soundness:** 3
**Presentation:** 4
**Contribution:** 3
**Rating:** 6
**Confidence:** 5

**Summary:**

The paper presents Dropsembles, a novel uncertainty quantification technique for neural implicit functions such as Neural Radiance Fields (NeRFs) and Occupancy Networks. This method aims to achieve the high accuracy and calibration of deep ensembles while significantly reducing computational overhead. By integrating Monte Carlo dropout with ensemble strategies, and leveraging Elastic Weight Consolidation (EWC), Dropsembles effectively handles distribution shifts and sparse, noisy data.

The primary focus is on addressing the challenges of 3D shape reconstruction from sparse and corrupted inputs. The method leverages high-quality, densely sampled synthetic datasets as shape priors to fill gaps and correct corruptions in the target data, even when distribution shifts occur due to noise and corruption. The premise is that these priors can guide the reconstruction process, ensuring more reliable outputs.

Initially, a Convolutional Occupancy Network is trained on high-quality synthetic data. During testing, the network is fine-tuned on individual sparse and noisy samples using EWC, which regularizes the adaptation process to prevent forgetting the original training data and to transfer prior knowledge effectively. This method involves creating ensembles through dropout, where binary masks generate multiple network instances. Each instance is then fine-tuned independently on the testing sample, forming an ensemble of networks initialized with correlated weights.

The paper validates the method through extensive experiments on toy datasets and low-resolution MRI segmentations of the lumbar spine. Results demonstrate that Dropsembles maintains accuracy and reliable uncertainty estimation and efficiently handles computational constraints and distribution shifts.

**Strengths:**

- The focus on uncertainty modeling in fine-tuned neural implicit functions addresses an underexplored area, contributing valuable insights into this challenging problem space.

- Dropsembles combines dropout and deep ensembles, offering a computationally efficient  training method for uncertainty estimation specifically tailored to neural implicit functions. This application of established uncertainty quantification and continual learning techniques to neural implicit functions is considered a novel contribution to neural implicit functions.

- The paper provides a rigorous experimental evaluation across synthetic and real-world spine datasets, effectively demonstrating that Dropsembles maintain prediction accuracy and uncertainty calibration under varying conditions.

- The method achieves significant reductions in computational costs during training compared to traditional deep ensembles, without compromising performance. However, computational demands during inference remain comparable to those of ensemble methods.

- The integration of Elastic Weight Consolidation (EWC) effectively mitigates the impact of distribution shifts, enhancing the model's robustness to noisy and sparse inputs.

- The approach is straightforward and versatile, capable of being integrated into various neural implicit networks and adapted to different task-specific training objectives, making it broadly applicable.

- The strategy of training on synthetic data while addressing distribution shifts between synthetic and real data is a pragmatic solution, particularly in scenarios where real-world data is scarce and noisy.

- The paper is clearly written, well-structured, and easy to follow.

**Weaknesses:**

- While the method shows promise in specific tasks like lumbar spine reconstruction, broader application beyond medical imaging is not thoroughly explored or validated.
- Despite reduced costs compared to deep ensembles, Dropsembles still require significant resources during fine-tuning, especially on high-resolution data.
- Although Dropsembles handle sparse inputs well, the effectiveness on highly varied real-world scenarios with extreme data corruption could have been examined more comprehensively.
- While Dropsembles generally outperform baselines, the improvements in some scenarios, particularly when using EWC, are not consistently significant across all metrics.

**Questions:**

- Could you provide more details on how the EWC regularization strength was selected? Is there a rationale beyond the empirical ablation studies, or could a more systematic method be developed?

- Have you tested Dropsembles on other types of neural implicit functions or datasets beyond those presented? How do you anticipate the method will perform in domains with significantly different characteristics?

- While the method reduces training costs, inference demands remain comparable to traditional deep ensembles. Are there potential strategies to also optimize inference efficiency?

- Given that synthetic data might not capture all real-world complexities, how do you ensure the generalization of the model to highly variable or unseen real-world data?

- Can you elaborate on the effectiveness of Dropsembles in handling extreme distribution shifts? Are there specific limitations or edge cases where the method struggles?

- How easily can Dropsembles be integrated into existing neural implicit networks? Are there any specific prerequisites or limitations that practitioners should be aware of?

---

> ### Author Response · Authors · 2024-11-27
> **General response on Weaknesses to Reviewer iePp**
>
> ## Broader application beyond medical imaging
>
> We appreciate the reviewer's comment and would like to refer to our *General Response* and the revised version of the manuscript where we provide new results on ShapeNet dataset.
>
> ## Resources during fine-tuning
>
> It is true that during fine-tuning on Dataset B, the cost is comparable to Ensembles, as multiple network instances are tuned independently. However, since the method is designed for an asymmetric transfer learning setting—where Dataset A is large and dense while Dataset B is sparse and small—the fine-tuning cost is relatively minor compared to the pretraining cost. While fine-tuning does represent a computational limitation, it is not significant in the context of the overall method.
>
> ## Effectiveness on highly varied real-world scenarios with extreme data corruption
>
> We would like to point out that our paper addresses a real-world clinical application characterized by a strong distribution shift. Specifically, we utilized synthetic data from an anatomical atlas as Dataset A to pretrain our network. For dataset B, we considered two increasingly complex scenarios:
> 1. **Controlled Corruption:** Dataset B was created by applying strong erosion and augmentation to simulate significant corruption, allowing us to generate ground truth for quantitative metrics.
> 2. **Real-World Data:** A dataset derived from MRI scans of real patients was used, where segmentations were produced by third-party neural networks. These segmentations were subjected to additional erosions and displayed a strong distribution shift from the synthetic atlas representation of an idealized human model to the real-world variability of actual patient data.
>
> To better illustrate the strong distribution shift in the lumbar spine application, we added Figures 12 and 13 with corresponding examples to the Appendix in the revised manuscript, clearly highlighting the differences between Dataset A and Dataset B.
>
>
> ## Performance of EWC
>
> We would like to note that in all datasets except the lumbar spine case (i.e., toy, MNIST, and ShapeNet), the improvements are significant as evident in metrics in Table 1. In the lumbar spine case, while the improvement is not substantial in the DSC score, this motivated the inclusion of the discussion section (496–501), where we address the limitations of standard metrics for such applications. Additionally, our claims were demonstrated in Figure 5 to further support our argument.
>
> We acknowledge that the numerical improvements on the lumbar spine dataset are subtle. However, we consider them meaningful because capturing small anatomical details is critical in real-world medical applications. These details are often challenging to reflect in metrics that are computed on high-resolution volumes and dominated by larger structures. We believe that designing metrics tailored to sensitive applications, such as medical imaging, is an important direction for future research.

---

> ### Author Response · Authors · 2024-11-28
> **Point by point response to questions to Reviewer iePp (part 1)**
>
> We sincerely thank the reviewer for taking the time to carefully evaluate the details of the proposed method. We hope that the responses below provide clear and satisfactory clarifications to the raised questions. Should there be any further concerns, we would be happy to address them.
>
> **- Could you provide more details on how the EWC regularization strength was selected? Is there a rationale beyond the empirical ablation studies, or could a more systematic method be developed?**
>
> The strength of EWC was selected via an ablation study. We observed a heuristic pattern for selecting the optimal strength for EWC, which depends on the average trade-off between the reconstruction/classification loss values and the EWC penalty. Since individual EWC coefficients are calculated from the gradient on Dataset A, their values are typically quite low. Consequently, a good strength value generally lies in the range of 1/10 to 1/1000 of the reconstruction loss and demonstrates the best performance.
>
> If the regularization strength is chosen too low, the EWC regularization will have no noticeable effect compared to the non-regularized version. Conversely, if the strength is set too high, the penalty will dominate the loss, hindering adaptation during fine-tuning and preventing the model from learning effectively from Dataset B.
>
> **- Have you tested Dropsembles on other types of neural implicit functions or datasets beyond those presented? How do you anticipate the method will perform in domains with significantly different characteristics?**
>
> Yes, in response to the reviewers' suggestions, we tested Dropsembles for modeling uncertainty in deep SDFs. Additionally, we explored modeling uncertainty in deep SDFs using sinusoidal activations. Details of this study can be found in the "General Response" and in the revised version of the manuscript.
>
> **- While the method reduces training costs, inference demands remain comparable to traditional deep ensembles. Are there potential strategies to also optimize inference efficiency?**
>
> Yes, inference efficiency could be optimized to a certain extent via a more efficient implementation. In the current implementation, we applied masking to remove the dropped units. By implementing network pruning, the number of active parameters used for matrix multiplications during inference could be reduced. However, this reduction depends on the dropout strength. In our study, we used moderate dropout probabilities of 0.2–0.3, so the improvement in inference speed would only be proportional to the dropout rate. For the dense decoder network used in this study, this would result in a 15–20% improvement in inference speed for a dropout probability of 0.2.
>
> **- How easily can Dropsembles be integrated into existing neural implicit networks? Are there any specific prerequisites or limitations that practitioners should be aware of?**
>
> Dropsembles are highly straightforward to integrate into any neural implicit network that supports dropout. The practical implementation involves using a standard dropout layer during training on Dataset A, which is replaced with randomly sampled, fixed masks for Dataset B in its simplest form. For a more advanced implementation, the fixed masks can be leveraged to prune the network fully, aligning the active parameters with the given mask to potentially optimize inference efficiency. Implementation of EWC is also straightforward and depends on computing the gradient of the network on dataset A. These weights can be precomputed and stored right at the end of the training stage on dataset A.

---

> ### Author Response · Authors · 2024-11-28
> **Point by point response to questions to Reviewer iePp (part 2)**
>
> **- Can you elaborate on the effectiveness of Dropsembles in handling extreme distribution shifts? Are there specific limitations or edge cases where the method struggles?**
>
> Handling strong distribution shifts is at the core of Dropsembles and serves as the key motivation for the design of our method:
>
> - **Separation of Encoder and Decoder:** By separating the encoder and decoder and introducing a fine-tuning stage, our method efficiently leverages the information available from sparse or corrupted data in Dataset B. The fine-tuning stage is especially crucial, as directly applying a network pretrained on Dataset A to Dataset B leads to poor generalization, as demonstrated by our metrics (see the Appendix in the revised version).
>
> - **EWC Regularization:** Another important aspect of Dropsembles is the use of EWC regularization, specifically chosen to mitigate overfitting (please refer to Appendix A and the General Response for further details). EWC regularization also is especially advantageous for the case of stron corruption, because it allows to not overfit to "mislabeled" samples induced by the corrution (e.g. values set to "background" due to erosion).
>
> While our method performs well in most scenarios, there is one edge case where it struggles. This issue arises due to the encoder design, rather than the decoder. When the input data is highly corrupted and large portions of the shape are missing or replaced with background values, the convolutional encoder produces a latent representation that corresponds to the "background" in these regions. Since these values are technically in-distribution, the decoder struggles to recognize them as areas of high uncertainty. This issue is not unique to our approach, but rather affects all benchmarks considered, including dropout and ensembles. We believe this limitation could potentially be mitigated by improving the encoder design, such as by using network architectures more suited to handling point clouds.
>
> **- Given that synthetic data might not capture all real-world complexities, how do you ensure the generalization of the model to highly variable or unseen real-world data?**
>
> This question is closely related to the previous one. The synthetic vs. real scenario is a real-world example of a strong distribution shift, which our paper addresses through the following strategies:
> 1. **Fine-Tuning:** To adapt the pretrained network to the real data.
> 2. **EWC Regularization:** To mitigate overfitting during fine-tuning by preserving the knowledge captured on the synthetic dataset.
> 3. **Modeling Uncertainty:** To assess the reliability of the fine-tuned instances.
>
> Our primary goal is to effectively leverage the pretrained network on the synthetic dataset to fill in the missing gaps in the real data, while still preserving the fine details unique to real-world examples. Uncertainty plays a crucial role in this context, as it helps track the balance between confidently fitting to the synthetic priors and identifying uncertain regions specific to the real-world data.

---

> ### Author Response · Authors · 2024-12-01
>
> Dear Reviewer iePp,
>
> As that the end of the discussion period is approaching, we would like to ask if you have any further concerns or questions, particularly as a follow-up to our response? If we have addressed all your concerns, we hope you will consider increasing your score.
>
> Thank you in advance!

---

### Official Review · Reviewer_6vSX · 2024-11-05

**Soundness:** 2
**Presentation:** 3
**Contribution:** 2
**Rating:** 6
**Confidence:** 4

**Summary:**

This paper introduces 'Dropsembles', an approach to uncertainty modeling in fine-tuned neural implicit functions. 'Dropsembles' combines dropout-based methods with ensemble learning to balance computational efficiency and robust uncertainty estimation.
The authors conduct experiments on synthetic anatomical datasets and MRI segmentation tasks, demonstrating that 'Dropsembles' maintains predictive accuracy comparable to deep ensembles while reducing computational demands.

**Strengths:**

Key contributions include:

(1) A method for modeling epistemic uncertainty in fine-tuned implicit functions.

(2) The application of Elastic Weight Consolidation (EWC) to address distribution shifts.

(3) Experimental validation across toy and real-world datasets.

**Weaknesses:**

(1) **Questionable motivation for EWC**:

**Lack of task distinction**: The dense and sparse datasets (Tasks A and B) are similar in content, which doesn’t align well with the typical use case for EWC in continual learning. EWC is usually applied to preserve knowledge across distinct tasks, so its application here appears unjustified or artificial.

(2) **Limited novelty in dropsembles**:

**Lack of specificity to NIR**:
Dropsembles combines dropout with ensembling in a straightforward way without tailoring it for neural implicit representations.
This method may not provide significant innovation for uncertainty estimation in 3D reconstruction, particularly if it doesn’t leverage the unique characteristics of implicit functions.

(3) **Insufficient baseline comparisons**:

Dropsembles is not compared to other standard uncertainty estimation techniques such as Bayesian neural networks, or other methods tailored to implicit representations [1]. This limits the evidence for Dropsembles’ effectiveness.

(4) **Limited dataset diversity**:

**Narrow generalizability**: The experiments are focused on synthetic anatomical data and sparse MRI segmentation, which may not fully demonstrate Dropsembles' robustness across various domains or types of data shifts.

References:

[1] Stochastic Neural Radiance Fields: Quantifying Uncertainty in Implicit 3D Representations, 3DV 2021

**Questions:**

**Q1**: Why is EWC necessary if Tasks A and B are similar?

Could you clarify why EWC is used in this context, given that Task A (dense dataset) and Task B (sparse/noisy dataset) do not appear to be significantly different? Is the intent to quantify uncertainty in a continual learning setting, or is EWC applied simply as a regularization tool?
What makes Dropsembles particularly suited to neural implicit representations?

**Q2**: Dropsembles seems like a standard dropout-ensemble approach. Are there specific adaptations that make it uniquely effective for implicit functions in 3D reconstructions, especially regarding spatial consistency or grid-based predictions?
Considering Dropsembles combines well-known methods, why weren’t additional baselines, like Bayesian neural networks or variational inference-based [1], tested to provide a clearer comparison?

**Q3**: How does Dropsembles handle spatial dependencies?

Neural implicit representations often require spatial coherence across grid or voxel predictions.
Does Dropsembles include any mechanism to maintain this spatial consistency in its uncertainty estimation, like [2]?

Could you discuss the limitations of Dropsembles in more detail?

**Q4**: Given that Dropsembles simply ensembles thinned models from dropout, what are its limitations in terms of computational cost, reliability, or adaptability to different 3D tasks?

**Minor**: The term "thinned network" isn’t a standard term in the field and may lead to confusion. It could be misinterpreted, as it doesn't clearly indicate that dropout is being applied.

**References**:

[1] Stochastic Neural Radiance Fields: Quantifying Uncertainty in Implicit 3D Representations, 3DV 2021

[2] Stochastic Segmentation Networks: Modelling Spatially Correlated Aleatoric Uncertainty, NeurIPS 2021

---

> ### Author Response · Authors · 2024-11-25
> **Response to Q1: Distribution Shift and EWC Application**
>
> Even the tasks A and be the same, the distribution shift between Tasks A and B is significant and systematic in our setting, which leads to a significant drop in the performance of a model trained on Dataset A and directly applied to Dataset B:
>
> 1. **Data Distribution Differences:**
> - Task A: Dense, high-quality synthetic data with complete geometric information
> - Task B: Sparse, corrupted real-world observations with missing information and noise
> - Different sampling mechanisms between synthetic and real data
> - Systematic differences in shape variations and feature distributions
>
> **Example:** In medical imaging, synthetic data from anatomical atlases cannot capture the full range of patient-specific variations. While an atlas may represent "standard" anatomy, real patient data exhibits unique pathologies, anatomical variations, and age/gender-related differences that are not represented in the synthetic training set. This creates a fundamental distribution shift when transferring from synthetic to real data.
>
> 2. **Relevance of EWC:**
>
> - EWC's core principle of preserving important parameters while allowing adaptation aligns perfectly with our scenario
> - Fisher information identifies parameters crucial for high-quality synthetic data performance
> - Allows controlled adaptation to real-world data without catastrophic forgetting of shape priors
>
> 3. **Novel Application:**
> - While originally developed for different tasks, EWC's mathematical framework is equally valid for handling distribution shifts
> - Our results validate this approach - EWC consistently improves both accuracy and uncertainty calibration across datasets
> - Demonstrates effective transfer of shape priors while adapting to real-world data characteristics
>
>
> We additionally refer to **Appendix A** where we explain and demonstrate that EWC is a principled way to handle the fundamental challenge of **distribution shift**.

---

> ### Author Response · Authors · 2024-11-26
> **Response to Q2: Comparing Uncertainty Estimation Approaches for Neural Implicit Representations**
>
> ## Bayesian Neural Networks
>
> Bayesian Neural Networks are indeed a desirable framework due to their ability to provide posterior distributions over model weights given observational data. However, their application to real-world datasets has been significantly hindered by their computational complexity and the need for non-trivial modifications to both architectures and training procedures. These challenges are well-documented, including in the specific domain of 3D modeling and estimation, as highlighted by the reviewer. For instance, the papers referenced by the reviewer, [1] (pg. 2, col. 2, par. 4) and [2] (pg. 2, par. 6), also acknowledge these limitations and, as a result, do not employ BNNs as baselines.
>
> ## S-NeRF: Variational Inference
>
> The S-NeRF paper suggested by the reviewer models epistemic uncertainty in NeRFs using variational inference (VI). While we agree that VI could be an interesting baseline, a computational comparison is infeasible in our work for several reasons. First, S-NeRF derives a VI framework specifically for NeRF optimization, which constitutes a significant contribution of their work. In contrast, our study does not include a direct NeRF benchmark, and to our knowledge, VI has not been demonstrated for modeling uncertainty in occupancy networks or SDFs, as considered in our paper. Applying VI to these tasks would require carefully designing a VI objective tailored to these models, which is beyond the scope of a simple comparison to existing baselines. However, we acknofledge the importance of an existence of such a baseline in NeRF modeling tasks and we added this discussion to the introduction of our paper.
>
> -----
>
> We appreciate the reviewer’s recognition of the value provided by these methods in improving uncertainty estimates for implicit representations. However, we would like to emphasize that the problems and scenarios addressed in these works differ from ours. Our approach specifically targets epistemic uncertainty under covariate shift, a challenge that is not directly tackled by [1] or [2].
>
> While it could be an interesting avenue for future work to explore complementing their methods with ours—for instance, by applying Elastic Weight Consolidation to [1] and [2]—this would require additional analysis to ensure computational efficiency. We believe such extensions fall outside the scope of the current manuscript, which focuses on foundational insights into our method.
>
> We hope this clarification highlights the distinct contributions of our work and its relevance to the problem at hand.

---

> ### Author Response · Authors · 2024-11-26
> **Response to Q3: How does Dropsembles handle spatial dependencies?**
>
> Dropsembles maintain spatial dependencies through two key mechanisms:
>
> 1. **Convolutional Encoder:**
>    We adopt the encoder-decoder architecture proposed in the Convolutional Occupancy Networks paper. The convolutional encoder, combined with bilinear upsampling, has been shown to maintain strong spatial consistency.
>
> 2. **Tuning of Thinned Networks:**
>    We would like to refer to "General Response" and the revised manuscript, where we added a detailed discussion on why Dropsembles is suitable for INR tasks, and in particular why vanilla dropout does not have this property.
>
> While both approaches handle spatial dependencies, they do so differently:
>
> **Stochastic Segmentations Networks's Approach:**
> - Models joint distribution over entire label maps using multivariate normal distribution in logit space
> - Directly captures pixel/voxel correlations through low-rank covariance matrix
> - Focuses on aleatoric uncertainty in single-domain segmentation
>
> **Our Approach:**
> - Maintains spatial consistency through:
>   - Network architecture's inherent spatial processing (convolutions/implicit functions)
>   - Dropout patterns that preserve architectural spatial dependencies
>   - EWC regularization that preserves learned shape priors
> - Handles epistemic uncertainty during fine-tuning
>
> Our method achieves spatial coherence by leveraging the network's existing spatial processing rather than explicitly modeling pixel correlations. This is evidenced by our coherent shape variations in the medical imaging results, where anatomical consistency is crucial, and additionally demonstrated in new ShapeNet results.

---

> ### Author Response · Authors · 2024-11-27
> **Response to Q4: discuss the limitations of Dropsembles in more detail**
>
> **Computational Cost:**
> As discussed in the paper and demonstrated in the metrics, the computational cost of Dropsembles falls between that of MC Dropout and Ensembles. During pretraining on Dataset A, Dropsembles incurs the same cost as MC Dropout, which is an advantage rather than a limitation. Compared to Ensembles, Dropsembles reduce the pretraining cost by a factor equal to the number of ensemble instances. During fine-tuning on Dataset B, the cost is comparable to Ensembles, as multiple network instances are tuned independently. However, since the method is designed for an asymmetric transfer learning setting—where Dataset A is large and dense while Dataset B is sparse and small—the fine-tuning cost is relatively minor compared to the pretraining cost. While fine-tuning does represent a computational limitation, it is not significant in the context of the overall method.
>
> **Computational Cost:**
> As discussed in the paper and demonstrated in the metrics, the computational cost of Dropsembles falls between that of MC Dropout and Ensembles. During pretraining on Dataset A, Dropsembles incurs the same cost as MC Dropout, which is an advantage rather than a limitation. Compared to Ensembles, Dropsembles reduce the pretraining cost by a factor equal to the number of ensemble instances. During fine-tuning on Dataset B, the cost is comparable to Ensembles, as multiple network instances are tuned independently. However, since the method is designed for an asymmetric transfer learning setting—where Dataset A is large and dense while Dataset B is sparse and small—the fine-tuning cost is relatively minor compared to the pretraining cost. While fine-tuning does represent a computational limitation, it is not significant in the context of the overall method.
>
> **Reliability:**
> In Appendix E (Figure 9), we provide results on the reliability of Dropsembles with varying numbers of thinned network instances. Similar to Ensembles, increasing the number of network instances improves both prediction accuracy and uncertainty estimates. However, this improvement comes at an additional computational cost. Notably, the performance gains from increasing the number of instances tend to saturate, making further computational expense unjustifiable in many cases.
>
> **Adaptability:**
> Dropsembles are straightforward to adapt to any 3D task that supports training with dropout. The main limitation lies in cases where dropout significantly degrades model performance, as not all architectures are equally robust to dropout. However, in this study, we tested Dropsembles with Convolutional Occupancy Networks, DeepSDF, and DeepSDF with SIREN activation functions, and observed strong performance across all tested architectures. Importantly, our work focuses on settings involving strong distribution shifts, such as synthetic-to-real generalization, which differ significantly from the prior applications that primarily consider in-distribution uncertainty.

---

> ### Author Response · Authors · 2024-11-27
> **General Response to Weaknesses**
>
> We thank the reviewer for their constructive feedback, which we have addressed in the revised manuscript. Specifically, we refer the reviewer to the *General Response* and revised version of the manuscript, where we provide detailed discussions on the specificity to NIR and dataset diversity comparison questions, along with new results on the ShapeNet dataset. The remaining weaknesses have been addressed point-by-point in our responses to each of the raised concerns.
>
> We sincerely appreciate the reviewer’s insightful comments and suggestions, which have greatly improved our manuscript. We have carefully addressed each point, conducted additional experiments, and provided analyses to strengthen our submission. We hope these revisions demonstrate that our work is ready for acceptance. Should there be any further questions, we are more than happy to address them.

---

> > ### Comment · Reviewer_6vSX · 2024-11-29
> >
> > Thank you for the detailed responses which addressed a large part of the raised issues with new experiments.
> > I raised my score to 'marginally above the acceptance threshold'.
> > This approach is simple, and kind of limited for epistemic uncertainty estimation under the domain shift scenario.
> > But it might raise discussion at ICLR.

---

### Author Response · Authors · 2024-11-25
**General response to all reviewers**

We thank the reviewers for their time and valuable feedback. We appreciate
the thorough comments which after addressing have improved our manuscript. Here we will summarize our response addressing common/essential concerns and then follow with point by point responses. We additionally provide a revised version of the manuscript, where changes are highlighted in "red".

Here we summarize the main points raised by the reviewers:

# Generalizability and additional results

## 1. Generalizability beyond medical shapes and occupancy networks

Upon reviewers' suggestion we provide additional experiments on **ShapeNet** dataset **with SDF** representations (details on this experiment could be found in the revised version on the manuscript):


| Method | mDICE [%] ↑ | mIoU [%] ↑ | ECE [%] ↓ |
|--------|-------------|-------------|--------|
| Dropout [trained only on A] | 78.4 ± 2.4 | 64.5 ± 3.2 | 31.66 ± 3.37 |
| Ensembles [trained only on A] | 76.4 ± 2.9 | 61.9 ± 3.7 | 36.33 ± 3.11 |
|--------|-------------|-------------|--------|
| Dropout | 82.8 ± 2.2 | 70.8 ± 3.3 | 19.31 ± 3.97 |
| Dropsembles | 83.5 ± 2.1 | 71.7 ± 3.2 | 19.51 ± 4.64 |
| Ensembles | 82.6 ± 2.2 | 70.5 ± 3.2 | 25.88 ± 3.27 |
| Dropout + EWC | 83.9 ± 2.4 | 72.3 ± 3.6 | **16.96** ± 3.81 |
| Dropsembles + EWC | **84.3** ± 2.1 | **72.9** ± 3.2 | 17.67 ± 4.45 |
| Ensembles + EWC | 83.4 ± 1.9 | 71.6 ± 2.8 | 24.59 ± 2.98 |


## 2.  Generalizability to other forms of INR

As pointed out by the reviewers, the INR community shifted to using activation functions beyond ReLU activations. Below the experimental results on ShapeNet dataset for **sine activation functions (SIREN)** [Sitzmann et al., 2020], demonstrating that Dropsembles can be efficiently applied to sine activation functions too.

| Method | mDICE [%] ↑ | mIoU [%] ↑ | ECE [%] ↓ |
|--------|-------------|-------------|--------|
| Dropout - SIREN [trained only on A] | 78.5 ± 3.3 | 64.8 ± 4.4 | 31.94 ± 3.89 |
| Ensembles - SIREN [trained only on A] | 75.9 ± 3.7 | 61.3 ± 4.8 | 36.44 ± 3.85 |
|--------|-------------|-------------|--------|
| Dropout | 80.2 ± 2.4 | 67.0 ± 3.3 | 27.99 ± 2.85 |
| Dropsembles | 83.8 ± 2.2 | 72.2 ± 3.2 | 21.04 ± 3.04 |
| Ensembles | 83.4 ± 2.5 | 71.6 ± 3.6 | 24.44 ± 3.03 |
| Dropout + EWC | 83.3 ± 2.2 | 71.5 ± 3.2 | 19.27 ± 2.74 |
| Dropsembles + EWC | **84.5** ± 2.3 | **73.2** ± 3.4 | **17.73** ± 3.05 |
| Ensembles + EWC | 80.9 ± 2.5 | 68.0 ± 3.5 | 28.95 ± 3.07 |



# Relevance of EWC
Addititonal resuls on ShapeNet from the tables above demonstrate consistent improvement across all methods with EWC regularization. In particular, in case of Dropsembles, EWC regularization brings additional +0.8% improvement over non-regularized version.

# Specificity of Dropsembles to uncertainty modeling in INR

## Dropsembels are better than vanilla Dropout for modeling uncertainty in INR
We would like to highlight the effectiveness and particular benefit of using Dropsembles versus vanila Dropout for INR. We do agree that Dropout is used in INR modeling for the purpose of regularization [Park et al. 2019] and is a commonly used method for modeling uncertainty in computer vision tasks [Kendall et al. 2017]. However, the *application of Dropout for the purpose of uncertainty modeling in INR*, to our knowledge, has not been explored and could benefit from a closer inspection.
We illustrated this crucial point in Figure 2(c-d), where we demonstrated that Dropout produced "speckle-type artifacts" in both reconstructions and uncertainties. To additionally support this point, we add Figure 11 in Appendix E where we demonstrate exactly the same behavior on ShapeNet dataset modeled with DeepSDF approach. From these figures it is apparent that the Dropout impairs the ability of the model to capture the true surface of the shape, producing a shape speckled with wholes and overall coarse texture. These "speckle-type" artifacts in Dropout predictions and uncertainty are specific for INR training procedure: during training, dropout is applied randomly at each sample, but compared to normal segmentation/regression tasks in computer vision (e.g. segmentation or depth estimation) in INR a "sample of data" contains a random subsample of shape 3d-coordinates. This in turn leads that every random set of coordinates is optimized with a random network instance of dropout and dropout instances are not coherent over 3D coordinates anymore. Naturally, Dropout implemented in such a way, does not translate to trustworthy uncertainty estimates. But, complementing Dropout with ensembels as we do in Dropsembels, mitigates this issue (as presented in Figures 2 and 11), producing a smooth shape representation. Hence, with Dropsembels we introduce an uncertainty modeling method that is tailored to the complexity and requirements of 3D representations. Additionally, this is supported by quantitative results in Tables 1, 2 and Appendix Tables 4.

---

### Meta-Review · Area_Chair_gPFe · 2024-12-23

**Metareview:**

The paper proposes a method called Dropsembles for uncertainty estimation for implicit neural representations. The method combines dropout with model ensembles with a learned prior by fine-tuning a pre-trained model using elastic weight consolidation. The reviewers commented on both the simplicity of the approach and the fact that it constituted a general-purpose approach not necessarily tailored to implicit neural representations. However, they appreciated the importance of the application and (especially after revision) the breadth of evaluation in terms of datasets. There was some concern raised about the absence of comparison to Bayesian neural networks.

The paper received two borderline accepts and one strong accept, with these reviewers all suggesting that the paper will be of interest from the ICLR community.

**Additional Comments On Reviewer Discussion:**

The paper benefited from the revision and I commend the authors for addressing reviewer feedback. Not too much back and forth between the authors or reviewers.

---

### Decision · Program_Chairs · 2025-01-22

Accept (Poster)